# DIFFERENTIALLY PRIVATE DIFFUSION MODELS

## ABSTRACT

While modern machine learning models rely on increasingly large training datasets, data is often limited in privacy-sensitive domains. Generative models trained with differential privacy (DP) on sensitive data can sidestep this challenge, providing access to synthetic data instead. However, training DP generative models is highly challenging due to the noise injected into training to enforce DP. We propose to leverage diffusion models (DMs), an emerging class of deep generative models, and introduce *Differentially Private Diffusion Models* (DPDMs), which enforce privacy using differentially private stochastic gradient descent (DP-SGD). We motivate why DP-SGD is well suited for training DPDMs, and thoroughly investigate the DM parameterization and the sampling algorithm, which turn out to be crucial ingredients in DPDMs. Furthermore, we propose *noise multiplicity*, a simple yet powerful modification of the DM training objective tailored to the DP setting to boost performance. We validate our novel DPDMs on widely-used image generation benchmarks and achieve state-of-the-art (SOTA) performance by large margins. For example, on MNIST we improve the SOTA FID from 48.4 to 5.01 and downstream classification accuracy from 83.2% to 98.1% for the privacy setting DP-$(\varepsilon{=}10, \delta{=}10^{-5})$. Moreover, on standard benchmarks, classifiers trained on DPDM-generated synthetic data perform on par with task-specific DP-SGD-trained classifiers, which has not been demonstrated before for DP generative models.

## 1 INTRODUCTION

Modern deep learning usually requires significant amounts of training data. However, sourcing large datasets in privacy-sensitive domains is often difficult. To circumvent this challenge, generative models trained on sensitive data can provide access to large synthetic data instead, which can be used flexibly to train downstream models. Unfortunately, typical overparameterized neural networks have been shown to provide little to no privacy to the data they have been trained on. For example, an adversary may be able to recover training images of deep classifiers using gradients of the networks (Yin et al., 2021) or reproduce training text sequences from large transformers (Carlini et al., 2021). Generative models may even overfit directly, generating data indistinguishable from the data they have been trained on. In fact, overfitting and privacy-leakage of generative models are more relevant than ever, considering recent works that train powerful photo-realistic image generators on large-scale Internet-scraped data (Rombach et al., 2021; Ramesh et al., 2022; Saharia et al., 2022).

To protect the privacy of training data, one may train their model using differential privacy (DP). DP is a rigorous privacy framework that applies to statistical queries (Dwork et al., 2006; 2014). In our case, this query corresponds to the training of a neural network using sensitive data. Differentially private stochastic gradient descent (DP-SGD) (Abadi et al., 2016) is the workhorse of DP training of neural networks. It preserves privacy by clipping and noising the parameter gradients during training. This leads to an inevitable trade-off between privacy and utility; for instance, small clipping constants and large noise injection result in very private models that may be of little practical use.

DP-SGD has, for example, been employed to train generative adversarial networks (GANs) (Frigerio et al., 2019; Torkzadehmahani et al., 2019; Xie et al., 2018), which are particularly susceptible to privacy-leakage (Webster et al., 2021). However, while GANs in the non-private setting can synthesize photo-realistic images (Brock et al., 2019; Karras et al., 2020b;a; 2021), their application in the private setting is challenging. GANs are difficult to optimize (Arjovsky & Bottou, 2017; Mescheder et al., 2018) and prone to mode collapse; both phenomena may be amplified during DP-SGD training.

Recently, Diffusion Models (DMs) have emerged as a powerful class of generative models (Song et al., 2021c; Ho et al., 2020; Sohl-Dickstein et al., 2015), demonstrating outstanding performance

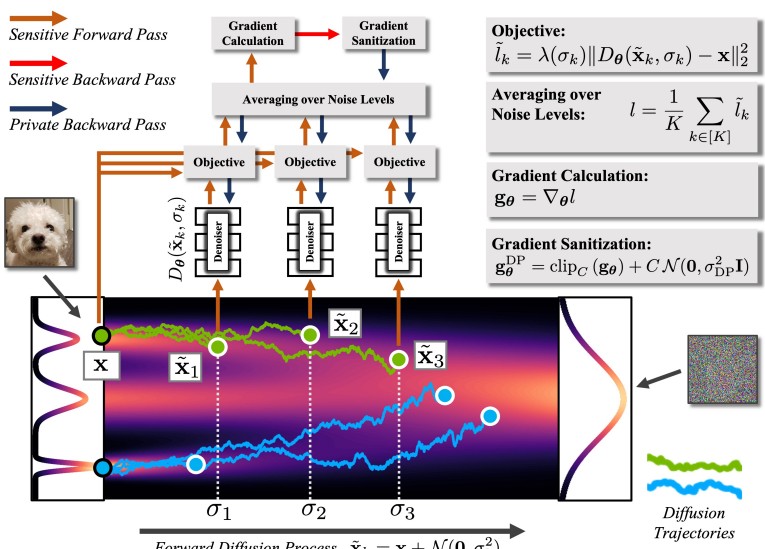

Figure 1: Information flow during training in our *Differentially Private Diffusion Model* (DPDM) for a single training sample in **green** (*i.e.* batchsize $B=1$, another sample shown in **blue**). We rely on DP-SGD to guarantee privacy and use *noise multiplicity*; here, $K=3$. The diffusion is visualized for a one-dim. toy distribution (marginal probabilities in **purple**); our main experiments use high-dim. images. Note that for brevity in the visualization we dropped the index $i$, which indicates the minibatch element in Eqs. (6) and (7).

in image synthesis (Ho et al., 2021; Nichol & Dhariwal, 2021; Dhariwal & Nichol, 2021; Rombach et al., 2021; Ramesh et al., 2022; Saharia et al., 2022). In DMs, a diffusion process gradually perturbs the data towards random noise, while a deep neural network learns to denoise. DMs stand out not only by high synthesis quality, but also sample diversity, and a simple and robust training objective. This makes them arguably well suited for training under DP perturbations. Moreover, generation in DMs corresponds to an iterative denoising process, breaking the difficult generation task into many small denoising steps that are individually simpler than the one-shot synthesis task performed by GANs and other traditional methods. In particular, the denoising neural network that is learnt in DMs and applied repeatedly at each synthesis step is less complex and smoother than the generator networks of one-shot methods, as we validate in experiments on toy data. Therefore, training of the denoising neural network is arguably less sensitive to gradient clipping and noise injection required for DP.

Based on these observations, we propose *Differentially Private Diffusion Models* (DPDMs), DMs trained with rigorous DP guarantees based on DP-SGD. We thoroughly study the DM parameterization and sampling algorithm, and tailor them to the DP setting. We find that the stochasticity in DM sampling, which is empirically known to be error-correcting (Karras et al., 2022), can be particularly helpful in DP-SGD training to obtain satisfactory perceptual output quality. We also propose *noise multiplicity*, where a single training data sample is re-used for training at multiple perturbation levels along the diffusion process (see Fig. 1). This simple yet powerful modification of the DM training objective improves learning at no additional privacy cost. We validate DPDMs on standard DP image generation tasks, and achieve state-of-the-art performance by large margins, both in terms of perceptual quality and performance of downstream classifiers trained on synthetically generated data from our models. For example, on MNIST we improve the state-of-the-art FID from 48.4 to 5.01 and downstream classification accuracy from 83.2% to 98.1% for the privacy setting DP-$(\varepsilon=10, \delta=10^{-5})$. We also find that classifiers trained on DPDM-generated synthetic data perform on par with task-specific DP-trained classifiers on standard benchmarks, which has not been demonstrated before for DP generative models.

In summary, we make the following contributions: **(i)** We carefully motivate training DMs with DP-SGD and introduce DPDMs, the first DMs trained under DP guarantees. **(ii)** We study DPDM parameterization, training setting and sampling in detail, and optimize it for the DP setup. **(iii)** We propose *noise multiplicity* to efficiently boost DPDM performance. **(iv)** Experimentally, we significantly surpass the state-of-the-art in DP synthesis on widely-studied image modeling benchmarks. **(v)** We demonstrate that classifiers trained on DPDM-generated data perform on par with task-specific DP-trained discriminative models. This implies a very high utility of the synthetic data generated by DPDMs, delivering on the promise of DP generative models as an effective data sharing medium. Finally, we hope that our work has implications for the literature on DMs, which are now routinely trained on ultra large-scale datasets of diverse origins.

## 2 BACKGROUND

### 2.1 DIFFUSION MODELS

We consider continuous-time DMs (Song et al., 2021c) and follow the presentation of Karras et al. (2022). Let $p_{\text{data}}(\mathbf{x})$ denote the data distribution and $p(\mathbf{x}; \sigma)$ the distribution obtained by adding i.i.d.

$\sigma^2$-variance Gaussian noise to the data distribution. For sufficiently large $\sigma_{\max}$, $p(\mathbf{x}; \sigma_{\max}^2)$ is almost indistinguishable from $\sigma_{\max}^2$-variance Gaussian noise. Capitalizing on this observation, DMs sample (high variance) Gaussian noise $\mathbf{x}_0 \sim \mathcal{N}\left(\mathbf{0}, \sigma_{\max}^2\right)$ and sequentially denoise $\mathbf{x}_0$ into $\mathbf{x}_i \sim p(\mathbf{x}_i; \sigma_i)$, $i \in [0, ..., M]$, with $\sigma_i < \sigma_{i-1}$ ($\sigma_0 = \sigma_{\max}$). If $\sigma_M = 0$, then $\mathbf{x}_0$ is distributed according to the data.

**Sampling.** In practice, the sequential denoising is often implemented through the simulation of the *Probability Flow* ordinary differential equation (ODE) (Song et al., 2021c)

$$d\mathbf{x} = -\dot{\sigma}(t)\sigma(t)\nabla_{\mathbf{x}} \log p(\mathbf{x}; \sigma(t)) \, dt, \tag{1}$$

where $\nabla_{\mathbf{x}} \log p(\mathbf{x}; \sigma)$ is the *score function* (Hyvärinen, 2005). The schedule $\sigma(t): [0, 1] \to \mathbb{R}_+$ is user-specified and $\dot{\sigma}(t)$ denotes the time derivative of $\sigma(t)$. Alternatively, we may also sample from a stochastic differential equation (SDE) (Song et al., 2021c; Karras et al., 2022):

$$d\mathbf{x} = \underbrace{-\dot{\sigma}(t)\sigma(t)\nabla_{\mathbf{x}} \log p(\mathbf{x}; \sigma(t)) \, dt}_{\text{Probability Flow ODE; see Eq. (1)}} - \underbrace{\beta(t)\sigma^2(t)\nabla_{\mathbf{x}} \log p(\mathbf{x}; \sigma(t)) \, dt + \sqrt{2\beta(t)}\sigma(t) \, d\omega_t}_{\text{Langevin diffusion component}}, \tag{2}$$

where $d\omega_t$ is the standard Wiener process. In principle, given initial samples $\mathbf{x}_0 \sim \mathcal{N}\left(\mathbf{0}, \sigma_{\max}^2\right)$, simulating either Probability Flow ODE or SDE produces samples from the same distribution. In practice, though, neither ODE nor SDE can be simulated exactly: Firstly, any numerical solver inevitably introduces discretization errors. Secondly, the score function is only accessible through a model $s_{\boldsymbol{\theta}}(\mathbf{x}; \sigma)$ that needs to be learned; replacing the score function with an imperfect model also introduces an error. Empirically, the ODE formulation has been used frequently to develop fast solvers (Song et al., 2021a; Zhang & Chen, 2022; Lu et al., 2022; Liu et al., 2022; Dockhorn et al., 2022a), whereas the SDE formulation often leads to higher quality samples (while requiring more steps) (Karras et al., 2022). One possible explanation for the latter observation is that the Langevin diffusion component in the SDE at any time during the synthesis process actively drives the process towards the desired marginal distribution $p(\mathbf{x}; \sigma)$, whereas errors accumulate in the ODE formulation, even when using many synthesis steps. In fact, it has been shown that as the score model $s_{\boldsymbol{\theta}}$ improves, the performance boost that can be obtained by an SDE solver diminishes (Karras et al., 2022). Finally, note that we are using classifier-free guidance (Ho & Salimans, 2021) to perform class-conditional sampling in this work. For details on classifier-free guidance and the numerical solvers for Eq. (1) and Eq. (2), we refer to App. C.3.

**Training.** DM training reduces to learning the score model $s_{\boldsymbol{\theta}}$. The model can, for example, be parameterized as $\nabla_{\mathbf{x}} \log p(\mathbf{x}; \sigma) \approx s_{\boldsymbol{\theta}} = (D_{\boldsymbol{\theta}}(\mathbf{x}; \sigma) - \mathbf{x})/\sigma^2$ (Karras et al., 2022), where $D_{\boldsymbol{\theta}}$ is a learnable *denoiser* that, given a noisy data point $\mathbf{x} + \mathbf{n}$, $\mathbf{x} \sim p_{\text{data}}(\mathbf{x})$, $\mathbf{n} \sim \mathcal{N}\left(\mathbf{0}, \sigma^2\right)$ and conditioned on the noise level $\sigma$, tries to predict the clean $\mathbf{x}$. The denoiser $D_{\boldsymbol{\theta}}$ can be trained by minimizing an $L_2$-loss

$$\arg\min_{\boldsymbol{\theta}} \mathbb{E}_{\mathbf{x} \sim p_{\text{data}}(\mathbf{x}), \sigma \sim p(\sigma), \mathbf{n} \sim \mathcal{N}(\mathbf{0}, \sigma^2)} \left[\lambda(\sigma) \| D_{\boldsymbol{\theta}}(\mathbf{x} + \mathbf{n}, \sigma) - \mathbf{x} \|_2^2 \right], \tag{3}$$

where $\lambda(\sigma): \mathbb{R}_+ \to \mathbb{R}_+$ is a weighting function. Previous works proposed various denoiser models $D_{\boldsymbol{\theta}}$, noise distributions $p(\sigma)$, and weightings $\lambda(\sigma)$. We refer to the triplet $(D_{\boldsymbol{\theta}}, p, \lambda)$ as the DM *config*. Here, we consider four such configs: *variance preserving* (VP) (Song et al., 2021c), *variance exploding* (VE) (Song et al., 2021c), **v**-prediction (Salimans & Ho, 2022), and the config introduced in Karras et al. (2022) (referred to as *Elucidate* in this work); App. C.1 for details.

## 2.2 DIFFERENTIAL PRIVACY

DP is a rigorous mathematical definition of privacy applied to statistical queries; in our work the queries correspond to the training of a neural network using sensitive training data. Informally, training is said to be DP, if, given the trained weights $\boldsymbol{\theta}$ of the network, an adversary cannot tell with certainty whether a particular data point was part of the training data. This degree of certainty is controlled by two positive parameters $\varepsilon$ and $\delta$: training becomes more private as $\varepsilon$ and $\delta$ decrease. Note, however, that there is an inherent trade-off between utility and privacy: very private models may be of little to no practical use. To guarantee a sufficient amount of privacy, as a rule of thumb, $\delta$ should not be larger than $1/N$, where $N$ is number of training points $\{\mathbf{x}_i\}_{i=1}^N$, and $\varepsilon$ should be a small constant. More formally, we refer to $(\varepsilon, \delta)$-DP defined as follows (Dwork et al., 2006):

**Definition 2.1.** (Differential Privacy) A randomized mechanism $\mathcal{M}: \mathcal{D} \to \mathcal{R}$ with domain $\mathcal{D}$ and range $\mathcal{R}$ satisfies $(\varepsilon, \delta)$-DP if for any two datasets $d, d' \in \mathcal{D}$ differing by at most one entry, and for any subset of outputs $S \subseteq \mathcal{R}$ it holds that

$$\mathbf{Pr}\left[\mathcal{M}(d) \in S\right] \leq e^{\varepsilon} \mathbf{Pr}\left[\mathcal{M}(d') \in S\right] + \delta. \tag{4}$$

**DP-SGD.** We require a DP algorithm that trains a neural network using sensitive data. The workhorse for this particular task is differentially private stochastic gradient descent (DP-SGD) (Abadi et al., 2016). DP-SGD is a modification of SGD for which per-sample-gradients are clipped and noise is added to the clipped gradients; the DP-SGD parameter updates are defined as follows

$$\boldsymbol{\theta} \leftarrow \boldsymbol{\theta} - \frac{\eta}{B} \left( \sum_{i \in \mathbb{B}} \texttt{clip}_C \left( \nabla_{\boldsymbol{\theta}} l_i(\boldsymbol{\theta}) \right) + C \mathbf{z} \right), \quad \mathbf{z} \sim \mathcal{N}(\mathbf{0}, \sigma_{\text{DP}}^2 \boldsymbol{I}), \tag{5}$$

where $\mathbb{B}$ is a $B$-sized subset of $\{1, \dots, N\}$ drawn uniformly at random, $l_i$ is the loss function for data point $\mathbf{x}_i$, $\eta$ is the learning rate, and the clipping function is $\texttt{clip}_C(\mathbf{g}) = \min \{1, C/\|\mathbf{g}\|_2\} \mathbf{g}$. DP-SGD can be adapted to other first-order optimizers, such as Adam (McMahan et al., 2018).

**Privacy Accounting.** According to the *Gaussian mechanism* (Dwork et al., 2014), a single DP-SGD update (Eq. (5)) satisfies $(\varepsilon, \delta)$-DP if $\sigma_{\text{DP}}^2 > 2 \log(1.25/\delta) C^2/\varepsilon^2$. Privacy accounting methods can be used to *compose* the privacy cost of multiple DP-SGD training updates and to determine the variance $\sigma_{\text{DP}}^2$ needed to satisfy $(\varepsilon, \delta)$-DP for a particular number of DP-SGD updates with clipping constant $C$ and subsampling rate $B/N$. Also see App. A.

# 3 DIFFERENTIALLY PRIVATE DIFFUSION MODELS

We propose DPDMs, DMs trained with rigorous DP guarantees based on DP-SGD. In Sec. 3.1, we discuss the motivation for using DMs for DP generative modeling. In Sec. 3.2, we then discuss training and methodological details as well as DM design choices, and we prove that DPDMs satisfy DP.

## 3.1 MOTIVATION

**(i) Objective function.** GANs have so far been the workhorse of DP generative modeling (see Sec. 4), even though they are generally difficult to optimize (Arjovsky & Bottou, 2017; Mescheder et al., 2018) due to their adversarial training and propensity to mode collapse. Both phenomena may be amplified during DP-SGD training. DMs, in contrast, have been shown to produce outputs as good or even better than GANs' (Dhariwal & Nichol, 2021), while being trained with a very simple regression-like $L_2$-loss (Eq. (3)), which makes them robust and scalable in practice. DMs are therefore arguably also well-suited for DP-SGD-based training and offer better stability under gradient clipping and noising than adversarial training frameworks.

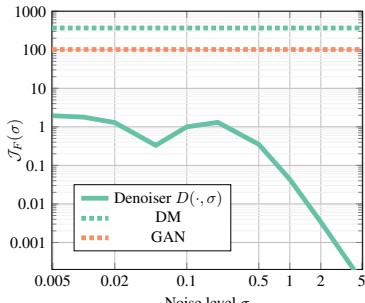

Figure 2: Frobenius norm of the Jacobian $\mathcal{J}_F(\sigma)$ of the denoiser $D(\cdot, \sigma)$ and constant Frobenius norms of the Jacobians $\mathcal{J}_F$ of the sampling functions defined by the DM and a GAN. App. D for experiment details.

**(ii) Sequential denoising.** In GANs and most other traditional generative modeling approaches, the generator directly learns the sampling function, i.e., the mapping of latents to synthesized samples, end-to-end. In contrast, the sampling function in DMs is defined through a sequential denoising process, breaking the difficult generation task into many small denoising steps which are individually less complex than the one-shot synthesis task performed by, for instance, a GAN generator. The denoiser neural network, the learnable component in DMs that is evaluated once per denoising step, is therefore simpler and smoother than the one-shot generator networks of other methods. We fit both a DM and a GAN to a two-dimensional toy distribution (mixture of Gaussians, see App. D) and empirically verify that the denoiser $D$ is indeed significantly less complex (quantified by the Frobenius norm of the Jacobian) than the generator learnt by the GAN and also than the end-to-end multi-step synthesis process (Probability Flow ODE) of the DM (see Fig. 2; we calculate denoiser $\mathcal{J}_F(\sigma)$ at varying noise levels $\sigma$). Generally, more complex functions require larger neural networks and are more difficult to learn. Note, however, that the $L_2$-norm of the noise added in the DP-SGD updates scales linearly with the number of parameters, and therefore smaller networks are generally preferred. Moreover, in DP-SGD training we only have a limited number of training iterations available until the privacy budget is depleted. Consequently, the fact that DMs require less complexity out of their neural networks than typical one-shot generation methods, while still being able to represent expressive generative models due to the iterative synthesis process, makes them likely well-suited for DP generative modeling with DP-SGD.

**(iii) Stochastic diffusion model sampling.** As discussed in Sec. 2.1, generating samples from DMs with stochastic sampling can perform better than deterministic sampling when the score model is not learned well. Since we replace gradient estimates in DP-SGD training with biased large variance

estimators, we cannot expect a perfectly accurate score model. In Sec. 5.2, we empirically show that stochastic sampling can in fact boost perceptual synthesis quality in DPDMs as measured by FID.

## 3.2 TRAINING DETAILS, DESIGN CHOICES, PRIVACY

The clipping and noising of the gradient estimates in DP-SGD (Eq. (5)) pose a major challenge for efficient optimization. Blindly reducing the added noise or increasing the clipping constant $C$ could be fatal, as it decreases the number of training iterations allowed within a certain $(\varepsilon, \delta)$-DP budget. Furthermore, as discussed the $L_2$-norm of the noise added in DP-SGD scales linearly to the number of parameters. Consequently, settings that work well for non-private DMs, such as relatively small batch sizes, a large number of training iterations, and heavily overparameterized models, may not work well for DPDMs. Below, we discuss how we propose to adjust DPDMs for successful DP-SGD training.

**Noise multiplicity.** Recall that the DM objective in Eq. (3) involves three expectations. As usual, the expectation with respect to the data distribution $p_{\text{data}}(\mathbf{x})$ is approximated using mini-batching. For non-private DMs, the expectations over $\sigma$ and $\mathbf{n}$ are generally approximated using a single Monte Carlo sample $(\sigma_i, \mathbf{n}_i) \sim p(\sigma)\mathcal{N}\left(\mathbf{0}, \sigma^2\right)$ per data point $\mathbf{x}_i$, resulting in the loss for training sample $i$

$$l_i = \lambda(\sigma_i)\|D_{\boldsymbol{\theta}}(\mathbf{x}_i + \mathbf{n}_i, \sigma_i) - \mathbf{x}_i\|_2^2. \tag{6}$$

The estimator $l_i$ is very noisy in practice. Non-private DMs counteract this by training for a large number of iterations in combination with an exponential moving average (EMA) of the trainable parameters $\boldsymbol{\theta}$ (Song & Ermon, 2020). When training DMs with DP-SGD, we incur a privacy cost for each iteration, and therefore prefer a small number of iterations. Furthermore, since the per-example gradient clipping as well as the noise injection induce additional variance, we would like our objective function to be less noisy than in the non-DP case. We achieve this by estimating the expectation over $\sigma$ and $\mathbf{n}$ using an average over $K$ noise samples, $\{(\sigma_{ik}, \mathbf{n}_{ik})\}_{k=1}^K \sim p(\sigma)\mathcal{N}\left(\mathbf{0}, \sigma^2\right)$ for each data point $\mathbf{x}_i$, replacing the non-private DM objective $l_i$ in Eq. (6) with

$$\tilde{l}_i = \frac{1}{K}\sum_{k=1}^K \lambda(\sigma_{ik})\|D_{\boldsymbol{\theta}}(\mathbf{x}_i + \mathbf{n}_{ik}, \sigma_{ik}) - \mathbf{x}_i\|_2^2. \tag{7}$$

Importantly, we show that this modification comes at *no* additional privacy cost (also see App. A). We call this simple yet powerful modification of the DM objective, which is tailored to the DP setup, *noise multiplicity*. The noise multiplicity mechanism is also highlighted in Fig. 1: the figure describes the information flow during training for a single training sample (i.e., batch size $B = 1$). Intuitively, the key is that we first create a relatively accurate low-variance gradient estimate by averaging over multiple noise samples before performing gradient sanitization in the backward pass via clipping and noising. Ideas similar to our noise multiplicity have recently been also used to train classifiers with DP-SGD, where multiple augmentations per image are used (De et al., 2022). We empirically showcase the benefit of noise multiplicity in Sec. 5.2.

**Neural networks sizes.** Current DMs are heavily overparameterized: For example, the current state-of-the-art image generation model (in terms of perceptual quality) on CIFAR-10 uses more than 100M parameters, despite the dataset consisting of only 50k training points (Karras et al., 2022). Using such heavily overparameterized models for DP-SGD training may not be effective because the $L_2$-norm of the noise added in the DP-SGD update scales linearly to the number of parameters. Furthermore, the per-example clipping operation of DP-SGD requires the computation of the loss gradient on each training example $\nabla_{\boldsymbol{\theta}}\tilde{l}_i$, rather than the minibatch gradient. In theory,

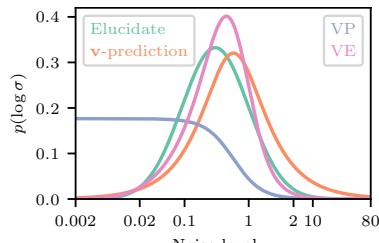

Figure 3: Noise level sampling for different DM configs; see App. C.1.

this increases the memory footprint by at least $\mathcal{O}(B)$; however, in practice the peak memory requirement is $\mathcal{O}(B^2)$ compared to non-private training (Yousefpour et al., 2021). On top of that, DP-SGD generally already relies on a significantly increased batch size, when compared to non-private training, to improve the privacy-utility trade-off. As a result, for both methodological as well as practical reasons, we train very small neural networks for DPDMs, when compared to their non-DP counterparts: our models on MNIST/Fashion-MNIST and CelebA have 1.75M and 1.80M parameters, respectively.

**Diffusion model config.** In addition to network size, we found the choice of DM config, i.e., denoiser parameterization $D_{\boldsymbol{\theta}}$, weighting function $\lambda(\sigma)$, and noise distribution $p(\sigma)$, to be important. In particular the latter is crucial to obtain strong results with DPDMs. In Fig. 3, we visualize the noise distributions of the four configs under consideration. We follow Karras et al. (2022) and plot the distribution

$p(\log \sigma)$ over the log-noise level. Especially for high privacy settings (small $\varepsilon$), we found it important to use distributions that give sufficiently much weight to larger $\sigma$, such as the distribution of **v**-prediction (Salimans & Ho, 2022). It is known that at large $\sigma$ the DM learns the global, coarse structure of the data, i.e., the low frequency content in the data (images, in our case). Learning global structure reasonably well is crucial to form visually coherent images that can also be used to train downstream models. This is relatively easy to achieve in the non-DP setting, due to the heavily smoothed diffused distribution at these high noise level. At high privacy levels, however, even training at such high noise levels can be challenging due to DP-SGD's gradient clipping and noising. We hypothesize that this is why it is beneficial to give relatively more weight to high noise levels when training in the DP setting. In Sec. 5.2, we empirically demonstrate the importance of the right choice of the DM config.

**DP-SGD settings.** Following De et al. (2022) we use very large batch sizes: 4096 on MNIST/Fashion-MNIST and 2048 on CelebA. Similar to previous works (De et al., 2022; Kurakin et al., 2022; Li et al., 2022), we found that small clipping constants $C$ work better than larger clipping norms; in particular, we found $C = 1$ to work well across our experiments. Decreasing $C$ even further had little effect; in contrast, increasing $C$ significantly worsened performance. Similar to non-private DMs, we use an EMA of the learnable parameters $\boldsymbol{\theta}$. Incidentally, this has recently been reported to also have a positive effect on DP-SGD training of classifiers by De et al. (2022).

**Privacy.** We formulate privacy protection under the Rényi Differential Privacy (RDP) (Mironov, 2017) framework (see Definition A.1), which can be converted to $(\epsilon, \delta)$-DP. For an algorithm for DPDM training with noise multiplicity see Alg. 1. For the sake of completeness we also formally prove the DP of DPDMs (DP of releasing sanitized training gradients $\tilde{G}_{batch}$):

---

**Algorithm 1** DPDM Training

**Input:** Private data set $d = \{\mathbf{x}_j\}_{j=1}^N$, subsampling rate $B/N$, DP noise scale $\sigma_{\text{DP}}$, clipping constant $C$, sampling function *Poisson Sample* (Alg. 2), denoiser $D_{\boldsymbol{\theta}}$ with initial parameters $\boldsymbol{\theta}$, noise distribution $p(\sigma)$, learning rate $\eta$, total steps $T$, noise multiplicity $K$, *Adam* (Kingma & Ba, 2015) optimizer
**Output:** Trained parameters $\boldsymbol{\theta}$
**for** $t = 1$ **to** $T$ **do**
    $\mathbb{B} \sim \textit{Poisson Sample}(N, B/N)$
    **for** $i \in \mathbb{B}$ **do**
        $\{(\sigma_{ik}, \mathbf{n}_{ik})\}_{k=1}^K \sim p(\sigma)\mathcal{N}(\mathbf{0}, \sigma^2)$
        $\tilde{l}_i = \frac{1}{K} \sum_{k=1}^K \lambda(\sigma_{ik}) \| D_{\boldsymbol{\theta}}(\mathbf{x}_i + \mathbf{n}_{ik}, \sigma_{ik}) - \mathbf{x}_i \|_2^2$
    **end for**
    $G_{batch} = \frac{1}{B} \sum_{i \in \mathbb{B}} \texttt{clip}_C \left( \nabla_{\boldsymbol{\theta}} \tilde{l}_i \right)$
    $\tilde{G}_{batch} = G_{batch} + (C/B)\mathbf{z}, \mathbf{z} \sim \mathcal{N}(\mathbf{0}, \sigma_{\text{DP}}^2)$
    $\boldsymbol{\theta} = \boldsymbol{\theta} - \eta * Adam(\tilde{G}_{batch})$
**end for**

---

**Theorem 1.** *For noise magnitude $\sigma_{\text{DP}}$, releasing $\tilde{G}_{batch}$ in Alg. 1 satisfies $\left(\alpha, \alpha/2\sigma_{\text{DP}}^2\right)$-RDP.*

The proof can be found in App. A. Note that the strength of DP protection is independent of the noise multiplicity, as discussed above. In practice, we construct mini-batches by *Poisson Sampling* (See Alg. 2) the training dataset for privacy amplification via sub-sampling (Mironov et al., 2019), and compute the overall privacy cost of training DPDM via RDP composition (Mironov, 2017).

## 4 RELATED WORK

In the DP generative learning literature, several works (Xie et al., 2018; Frigerio et al., 2019; Torkzadehmahani et al., 2019; Chen et al., 2020) have explored applying DP-SGD (Abadi et al., 2016) to GANs, while others (Yoon et al., 2019; Long et al., 2019; Wang et al., 2021) train GANs under the PATE (Papernot et al., 2018) framework, which distills private teacher models (discriminators) into a public student (generator) model. Apart from GANs, Acs et al. (2018) train variational autoencoders on DP-sanitized data clusters, and Cao et al. (2021) use the Sinkhorn divergence and DP-SGD.

DP-MERF (Harder et al., 2021) was the first work to perform one-shot privatization on the data, followed by non-private learning. It uses differentially private random Fourier features to construct a Maximum Mean Discrepancy loss, which is then minimized by a generative model. PEARL (Liew et al., 2022) instead minimizes an empirical characteristic function, also based on Fourier features. DP-MEPF (Harder et al., 2022) extends DP-MERF to the mixed public-private setting with pre-trained feature extractors. While these approaches are efficient in the high-privacy/small dataset regime, they are limited in expressivity by the data statistics that can be extracted during one-shot privatization. As a result, the performance of these methods does not scale well in the low-privacy/large dataset regime.

In our experimental comparisons, we excluded Takagi et al. (2021) and Chen et al. (2022) due to concerns regarding their privacy guarantees. The privacy analysis of Takagi et al. (2021) relies on the Wishart mechanism, which has been retracted due to privacy leakage (Sarwate, 2017). Chen

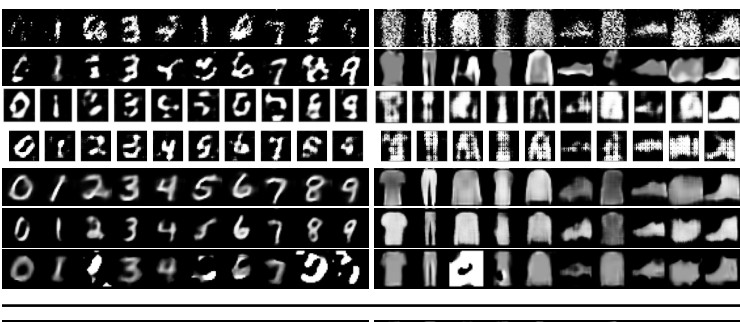

Figure 4: MNIST and Fashion-MNIST images generated by **DP-CGAN** (*1st row*), **DP-MERF** (*2nd row*), **Datalens** (*3rd row*), **G-PATE** (*4th row*), **GS-WGAN** (*5th row*), **DP-Sinkhorn** (*6th row*), **PEARL** (*7th row*) and our **DPDM** (*8th row*). The DP privacy setting is $\varepsilon=10$. Please see App. E.5 for more samples.

et al. (2022) attempt to train a score-based model while guaranteeing differential privacy through a data-dependent randomized response mechanism. In App. B, we prove why their proposed mechanism leaks privacy, and further discuss other sources of privacy leakage.

Our DPDM relies on DP-SGD (Abadi et al., 2016) to enforce DP guarantees. DP-SGD has also been used to train DP classifiers (Dörmann et al., 2021; Tramer & Boneh, 2021; Kurakin et al., 2022). Recently, De et al. (2022) demonstrated how to train very large discriminative models with DP-SGD and proposed augmentation multiplicity, which is related to our noise multiplicity, as discussed in Sec. 3.2. Furthrmore, DP-SGD has been utilized to train and fine-tune large language models (Anil et al., 2021; Li et al., 2022; Yu et al., 2022), to protect sensitive training data in the medical domain (Ziller et al., 2021a;b; Balelli et al., 2022), and to obscure geo-spatial location information (Zeighami et al., 2022).

Our work builds on DMs and score-based generative models (Sohl-Dickstein et al., 2015; Song et al., 2021c; Ho et al., 2020). DMs have been used prominently for image synthesis (Ho et al., 2021; Nichol & Dhariwal, 2021; Dhariwal & Nichol, 2021; Rombach et al., 2021; Ramesh et al., 2022; Saharia et al., 2022) and other image modeling tasks (Meng et al., 2021; Saharia et al., 2021a;b; Li et al., 2021; Sasaki et al., 2021; Kawar et al., 2022). They have also found applications in other areas, for instance in audio and speech generation (Chen et al., 2021; Kong et al., 2021; Jeong et al., 2021) and 3D synthesis (Luo & Hu, 2021; Zhou et al., 2021; Zeng et al., 2022). Methodologically, DMs have been adapted, for example, for fast sampling (Jolicoeur-Martineau et al., 2021; Song et al., 2021a; Salimans & Ho, 2022; Dockhorn et al., 2022b; Xiao et al., 2022; Watson et al., 2022; Dockhorn et al., 2022a) and maximum likelihood training (Song et al., 2021b; Kingma et al., 2021; Vahdat et al., 2021). To the best of our knowledge, we are the first to train DMs under differential privacy guarantees.

## 5 EXPERIMENTS

**Datasets.** We focus on image synthesis and use MNIST (LeCun et al., 2010), Fashion-MNIST (Xiao et al., 2017) (both 28x28 resolution), and CelebA (Liu et al., 2015) (center-cropped; downsampled to 32x32 resolution). These three datasets are widely used in the DP generative modeling literature as standard benchmarks. They contain 50k, 50k, and 162k training images, respectively.

**Architectures.** We implement the neural networks of DPDMs using the DDPM++ architecture (Song et al., 2021c). For class-conditioning, we add a learned class-embedding. See App. C.2 for details.

**Evaluation.** We measure sample quality via Fréchet Inception Distance (FID) (Heusel et al., 2017). On MNIST and Fashion-MNIST, we also assess utility of class-labeled generated data by training classifiers on synthesized samples and compute class prediction accuracy on real data. As is standard practice, we consider logistic regression (Log Reg), MLP, and CNN classifiers; see App. E.1 for details.

**Sampling.** We generate samples from DPDM using (stochastic) DDIM (Song et al., 2021c) and the Churn sampler introduced in (Karras et al., 2022). See App. C.3 for details and pseudocode.

**Privacy implementation:** We implement DPDMs in PyTorch (Paszke et al., 2019) and use Opacus (Yousefpour et al., 2021), a DP-SGD library in PyTorch, for training and privacy accounting. We use $\delta=10^{-5}$ for MNIST and Fashion-MNIST, and $\delta=10^{-6}$ for CelebA. These values are standard (Cao et al., 2021) and chosen such that $\delta$ is smaller than the reciprocal of the number of training images. Similar to existing DP generative modeling work, we do not account for the (small) privacy cost of hyperparameter tuning. However, training and sampling is very robust with regards to hyperparameters, which makes DPDMs an ideal candidate for real privacy-critical situations; see App. C.4.

### 5.1 MAIN RESULTS

**Class-conditional gray scale image generation.** For MNIST and Fashion-MNIST, we train models

Table 1: Class-conditional DP image generation performance (MNIST & Fashion-MNIST). For PEARL (Liew et al., 2022), we train models and compute metrics ourselves (App. E.1). All other results taken from the literature. DP-MEPF (†) uses additional public data for training (only included for completeness).

| Method | DP-$\varepsilon$ | MNIST | | | | Fashion-MNIST | | | |
|---|---|---|---|---|---|---|---|---|---|
| | | FID | Acc (%) | | | FID | Acc (%) | | |
| | | | Log Reg | MLP | CNN | | Log Reg | MLP | CNN |
| DPDM (FID) (ours) | 0.2 | **61.9** | 65.3 | 65.8 | 71.9 | **78.4** | 53.6 | 55.3 | 57.0 |
| DPDM (Acc) (ours) | 0.2 | 104 | **81.0** | **81.7** | **86.3** | 128 | **70.4** | **71.3** | **72.3** |
| PEARL (Liew et al., 2022) | 0.2 | 133 | 76.2 | 77.1 | 77.6 | 160 | 70.0 | 70.8 | 68.0 |
| DPDM (FID) (ours) | 1 | **23.4** | 83.8 | 87.0 | 93.4 | **37.8** | 71.5 | 71.7 | 73.6 |
| DPDM (Acc) (ours) | 1 | 35.5 | **86.7** | **91.6** | **95.3** | 51.4 | **76.3** | **76.9** | **79.4** |
| PEARL (Liew et al., 2022) | 1 | 121 | 76.0 | 79.6 | 78.2 | 109 | 74.4 | 74.0 | 68.3 |
| DPDM (FID) (ours) | 10 | **5.01** | 90.5 | 94.6 | 97.3 | **18.6** | 80.4 | 81.1 | 84.9 |
| DPDM (Acc) (ours) | 10 | 6.65 | **90.8** | **94.8** | **98.1** | 19.1 | **81.1** | **83.0** | **86.2** |
| PEARL (Liew et al., 2022) | 10 | 116 | 76.5 | 78.3 | 78.8 | 102 | 72.6 | 73.2 | 64.9 |
| DP-Sinkhorn (Cao et al., 2021) | 10 | 48.4 | 82.8 | 82.7 | 83.2 | 128.3 | 75.1 | 74.6 | 71.1 |
| G-PATE (Long et al., 2019) | 10 | 150.62 | - | - | 80.92 | 171.90 | - | - | 69.34 |
| DP-CGAN (Torkzadehmahani et al., 2019) | 10 | 179.2 | 60 | 60 | 63 | 243.8 | 51 | 50 | 46 |
| DataLens (Wang et al., 2021) | 10 | 173.5 | - | - | 80.66 | 167.7 | - | - | 70.61 |
| DP-MERF (Harder et al., 2021) | 10 | 116.3 | 79.4 | 78.3 | 82.1 | 132.6 | 75.5 | 74.5 | 75.4 |
| GS-WGAN (Chen et al., 2020) | 10 | 61.3 | 79 | 79 | 80 | 131.3 | 68 | 65 | 65 |
| DP-MEPF ($\phi_1$) (Harder et al., 2022) (†) | 0.2 | - | 72.1 | 77.1 | - | - | 71.7 | 69.0 | - |
| DP-MEPF ($\phi_1, \phi_2$) (Harder et al., 2022) (†) | 0.2 | - | 75.8 | 79.9 | - | - | 72.5 | 70.4 | - |
| DP-MEPF ($\phi_1$) (Harder et al., 2022) (†) | 1 | - | 79.0 | 87.5 | - | - | 76.2 | 75.0 | - |
| DP-MEPF ($\phi_1, \phi_2$) (Harder et al., 2022) (†) | 1 | - | 82.5 | 89.3 | - | - | 75.4 | 74.7 | - |
| DP-MEPF ($\phi_1$) (Harder et al., 2022) (†) | 10 | - | 80.8 | 88.8 | - | - | 75.5 | 75.5 | - |
| DP-MEPF ($\phi_1, \phi_2$) (Harder et al., 2022) (†) | 10 | - | 83.4 | 89.8 | - | - | 75.7 | 76.0 | - |

Table 2: Class prediction accuracy on real test data. DP-SGD: Classifiers trained directly with DP-SGD and real training data. DPDM: Classifiers trained non-privately on synthesized data from DP-SGD-trained DPDMs.

| DP-$\varepsilon$ | MNIST | | | | | | Fashion-MNIST | | | | | |
|---|---|---|---|---|---|---|---|---|---|---|---|---|
| | Log Reg | | MLP | | CNN | | Log Reg | | MLP | | CNN | |
| | DP-SGD | DPDM | DP-SGD | DPDM | DP-SGD | DPDM | DP-SGD | DPDM | DP-SGD | DPDM | DP-SGD | DPDM |
| 0.2 | **83.8** | 81.0 | **82.0** | 81.7 | 69.9 | **86.3** | **74.8** | 70.4 | **73.9** | 71.3 | 59.5 | **72.3** |
| 1 | **89.1** | 86.7 | 89.6 | **91.6** | 88.2 | **95.3** | **79.6** | 76.3 | **79.6** | 76.9 | 70.5 | **79.4** |
| 10 | **91.6** | 90.8 | 92.9 | **94.8** | 96.4 | **98.1** | **83.3** | 81.1 | **83.9** | 83.0 | 77.1 | **86.2** |

for three privacy settings: $\varepsilon = \{0.2, 1, 10\}$ (Tab. 1). Informally, the three settings provide high, moderate, and low amounts of privacy, respectively. The DPDMs use the **v**-prediction DM config (Salimans & Ho, 2022) for $\varepsilon = 0.2$ and the Elucidate DM config (Karras et al., 2022) for $\varepsilon = \{1, 10\}$; see Sec. 5.2. We use the Churn sampler (Karras et al., 2022): the two settings (FID) and (Acc) are based on the same DM, differing only in sampler setting; see Tab. 14 and Tab. 15 for all sampler settings.

DPDMs outperform all other existing models for all privacy settings and all metrics by large margins (see Tab. 1). Interestingly, DPDM also outperforms DP-MEPF (Harder et al., 2022), a method which is trained on additional public data, in 22 out of 24 setups. Generated samples for $\varepsilon = 10$ are shown in Fig. 4. Visually, DPDM's samples appear to be of significantly higher quality than the baselines'.

**Comparison to DP-SGD-trained classifiers.** Is it better to train a task-specific private classifier with DP-SGD directly, or can a non-private classifier trained on DPDM's synthethized data perform as well on downstream tasks? To answer this question, we train private classifiers with DP-SGD on real (training) data and compare them to our classifiers learnt using DPDM-synthesized data (details in App. E.3). For a fair comparison, we are using the same architectures that we have already been using in our main experiments to quantify downstream classification accuracy (results in Tab. 2; we test on real (test) data). While direct DP-SGD training on real data outperforms the DPDM downstream classifier for logistic regression in all six setups (in line with empirical findings that it is easier to train classifiers with

Table 3: Noise multiplicity ablation on MNIST for $\varepsilon = 1$. See Tab. 11 for extended results.

| K | FID | CNN-Acc (%) |
|---|---|---|
| 1 | 76.9 | 91.7 |
| 2 | 60.1 | 93.1 |
| 4 | 57.1 | 92.8 |
| 8 | 44.8 | 94.1 |
| 16 | 36.9 | 94.2 |
| 32 | 34.8 | 94.4 |

few parameters than large ones with DP-SGD (Tramer & Boneh, 2021)), CNN classifiers trained on DPDM's synthetic data generally outperform DP-SGD-trained classifiers. These results imply a very high utility of the synthetic data generated by DPDMs, demonstrating that DPDMs can potentially be used as an effective, privacy-preserving data sharing medium in practice. In fact, this approach is beneficial over training task-specific models with DP-SGD, because a user can generate as much data from DPDMs as they desire for various downstream applications without further privacy implications. To the best of our knowledge, it has not been demonstrated before in the DP generative modeling literature that (image) data generated by DP generative models can be used to train discriminative models on-par with directly DP-SGD-trained task-specific models.

**Unconditional color image generation.** On CelebA, we train models for $\varepsilon = \{1, 10\}$ (Tab. 4 & Fig. 5). The two DPDMs use the Elucidate config (Karras et al., 2022) as well as the Churn sampler; see Tab. 14. For $\varepsilon = 10$, DPDM again outperforms existing methods by a significant margin. DPDM's synthesized images appear much more diverse and vivid than the baselines' samples.

Table 4: (*bottom*) Unconditional CelebA generative performance. G-PATE and DataLens (†) use $\delta = 10^{-5}$ (less privacy) and model images at 64x64 resolution.

| Method | DP-$\varepsilon$ | FID |
|---|---|---|
| DPDM (*ours*) | 1 | 71.8 |
| DPDM (*ours*) | 10 | **21.1** |
| DP-Sinkhorn (Cao et al., 2021) | 10 | 189.5 |
| DP-MERF (Harder et al., 2021) | 10 | 274.0 |
| G-PATE (Long et al., 2019) (†) | 10 | 305.92 |
| DataLens (Wang et al., 2021) (†) | 10 | 320.8 |

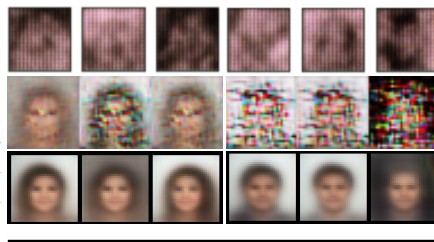

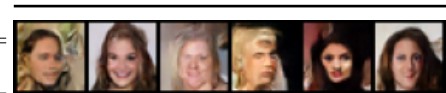

Figure 5: CelebA images generated by **DataLens** (*1st row*), **DP-MEPF** (*2nd row*), **DP-Sinkhorn** (*3rd row*), and our **DPDM** (*4th row*) for DP-$\varepsilon$=10. More samples in App. E.5.

## 5.2 ABLATION STUDIES

**Noise multiplicity.** Tab. 3 shows results for DPDMs trained with different noise multiplicity $K$. As expected, increasing $K$ leads to a general trend of improving performance; however, the metrics start to plateau at around $K$=32.

**Diffusion model config.** We train DPDMs with different DM configs (see App. C.1). VP- and VE-based models (Song et al., 2021c) perform poorly for all settings, while for $\varepsilon$=0.2 **v**-prediction significantly outperforms the Elucidate DM config on MNIST (Tab. 5).

Table 5: DM config ablation on MNIST for $\varepsilon$=0.2. See Tab. 12 for extended results.

| DM config | FID | CNN-Acc (%) |
|---|---|---|
| VP (Song et al., 2021c) | 197 | 24.2 |
| VE (Song et al., 2021c) | 171 | 13.9 |
| v-prediction (Salimans & Ho, 2022) | **97.8** | **84.4** |
| Elucidate (Karras et al., 2022) | 119 | 49.2 |

On Fashion-MNIST, the advantage is less significant (extended Tab. 12). For $\varepsilon$={1, 10}, the Elucidate DM config performs better than **v**-prediction. Note that the denoiser parameterization for these configs is almost identical and their main difference is the noise distribution $p(\sigma)$ (Fig. 3). As discussed in Sec. 3.2, oversampling large noise levels $\sigma$ is expected to be especially important for the large privacy setting (small $\varepsilon$), which is validated by our ablation.

**Sampling.** Tab. 6 shows results for different samplers: deterministic and stochastic DDIM (Song et al., 2021a) as well as the Churn sampler (tuned for high FID scores and downstream accuracy). Stochastic sampling is crucial to obtain good perceptual quality, as measured by FID (see poor performance of deterministic DDIM), while it is less important for downstream accuracy. We hypothesize that FID better captures image details that require a sufficiently accurate synthesis process. As discussed in Secs. 2.1 and 3.1, stochastic sampling can help with that and therefore is particularly important in DP-SGD-trained DMs. We also observe that the advantage of the Churn sampler compared to stochastic DDIM becomes less significant as $\varepsilon$ increases. Moreover, in particular for $\varepsilon$=0.2 the FID-adjusted Churn sampler performs poorly on downstream accuracy. This is arguably because its settings sacrifice sample diversity, which downstream accuracy usually benefits from, in favor of synthesis quality (also see samples in App. E.5).

## 6 CONCLUSIONS

We proposed *Differentially Private Diffusion Models* (DPDMs), which use DP-SGD to enforce DP guarantees during DM training. DMs are strong candidates for DP generative learning due to their robust training objective and intrinsically less complex denoising neural networks. We perform an in-depth analysis of the ideal DPDM parametrization and sampling strategy and introduce noise multiplicity to boost synthesis quality. DPDMs achieve state-of-the-art performance in common DP image generation benchmarks. Furthermore, downstream classifiers trained with DPDM-generated synthetic data perform on-par with task-specific discriminative models trained

Table 6: Diffusion sampler comparison on MNIST (see Tab. 13 for results on Fashion-MNIST). We compare the Churn sampler (Karras et al., 2022) to stochastic and deterministic DDIM (Song et al., 2021a).

| Sampler | DP-$\varepsilon$ | FID | Acc (%) | | |
|---|---|---|---|---|---|
| | | | Log Reg | MLP | CNN |
| Churn (FID) | 0.2 | **61.9** | 65.3 | 65.8 | 71.9 |
| Churn (Acc) | 0.2 | 104 | 81.0 | 81.7 | **86.3** |
| Stochastic DDIM | 0.2 | 97.8 | 80.2 | 81.3 | 84.4 |
| Deterministic DDIM | 0.2 | 120 | **81.3** | 82.1 | 84.8 |
| Churn (FID) | 1 | **23.4** | 83.8 | 87.0 | 93.4 |
| Churn (Acc) | 1 | 35.5 | **86.7** | 91.6 | **95.3** |
| Stochastic DDIM | 1 | 34.2 | 86.2 | 90.1 | 94.9 |
| Deterministic DDIM | 1 | 50.4 | 85.7 | **91.8** | 94.9 |
| Churn (FID) | 10 | **5.01** | 90.5 | 94.6 | 97.3 |
| Churn (Acc) | 10 | 6.65 | **90.8** | 94.8 | **98.1** |
| Stochastic DDIM | 10 | 6.13 | 90.4 | 94.6 | 97.5 |
| Deterministic DDIM | 10 | 10.9 | 90.5 | **95.2** | 97.7 |

with DP-SGD directly. Based on our promising results, we conclude that DMs are an ideal generative modeling framework for DP generative learning. We hope that DPDMs can grow into a practical tool for effective data sharing in the form of a generative model that can produce synthetic but useful data, while preserving the privacy of the generative model's original training data. Moreover, we believe that advancing DM-based DP generative modeling is a pressing topic, considering the extremely fast progress of DM-based large-scale photo-realistic image generation systems (Rombach et al., 2021; Saharia et al., 2022; Ramesh et al., 2022). As future directions we envision applying our DPDM approach during training of such large image generation DMs, as well as applying DPDMs to other types of data.

## 7 ETHICS AND REPRODUCIBILITY

Our work improves the state-of-the-art in differentially private generative modeling and we validate our proposed DPDMs on image synthesis benchmarks. Generative modeling of images has promising applications, for example for digital content creation and artistic expression (Bailey, 2020), but it can in principle also be used for malicious purposes (Vaccari & Chadwick, 2020; Mirsky & Lee, 2021; Nguyen et al., 2021). However, differentially private image generation methods, including our DPDM, are currently not able to produce photo-realistic content, which makes such abuse unlikely.

As discussed in Sec. 1, a severe issue in modern generative models is that they can easily overfit to the data distribution, thereby closely reproducing training samples and leaking privacy of the training data. Our DPDMs aim to rigorously address such problems via the well-established DP framework and fundamentally protect the privacy of the training data and prevent overfitting to individual data samples. This is especially important when training generative models on diverse and privacy-sensitive data. Therefore, DPDMs can potentially act as an effective medium for data sharing without needing to worry about data privacy, which we hope will benefit the broader machine learning community. Note, however, that although DPDM provides privacy protection in generative learning, information about individuals cannot be eliminated entirely, as no useful model can be learned under DP-$(\varepsilon{=}0, \delta{=}0)$. This should be communicated clearly to dataset participants.

To aid reproducibility of the results and methods presented in our paper, we will make source code to reproduce all quantitative and qualitative results of the paper publicly available, including detailed instructions. Moreover, all training details and hyperparameters are already described in detail in the Appendix, in particular in App. C.

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

CONTENTS

## A   Differential Privacy and Proof of Theorem 1

In this section, we provide a short proof that the gradients released by the Gaussian mechanism in DPDM are DP. By DP, we are specifically refering to the $(\varepsilon, \delta)$-DP as defined in definition 2.1, which approximates $(\varepsilon)$-DP. For completeness, we state the definition of Rényi Differential Privacy (RDP) (Mironov, 2017):

**Definition A.1.** (Rényi Differential Privacy) A randomized mechanism $\mathcal{M} : \mathcal{D} \to \mathcal{R}$ with domain $\mathcal{D}$ and range $\mathcal{R}$ satisfies $(\alpha, \epsilon)$-RDP if for any adjacent $d, d' \in \mathcal{D}$:

$$D_\alpha(\mathcal{M}(d)|\mathcal{M}(d')) \le \epsilon, \tag{8}$$

where $D_\alpha$ is the Rényi divergence of order $\alpha$.

Gaussian mechanism can provide RDP according to the following theorem:

**Theorem 2.** (RDP Gaussian mechanism (Mironov, 2017)) For query function $f$ with Sensitivity $S = \max_{d,d} ||f(d) - f(d')||_2$, the mechanism that releases $f(d) + \mathcal{N}(0, \sigma_{\mathrm{DP}}^2)$ satisfies $\left(\alpha, \alpha S^2/(2\sigma^2)\right)$-RDP.

Note that any $\mathcal{M}$ that satisfies $(\alpha, \epsilon)$-RDP also satisfies $(\epsilon + \frac{\log 1/\delta}{\alpha - 1}, \delta)$-DP.

We slightly deviate from the notation used in the main text to make the dependency of variables on input data explicit. Recall from the main text that the per-data point loss is computed as an average over $K$ noise samples:

$$\tilde{l}_i = \frac{1}{K} \sum_{k=1}^{K} \lambda(\sigma_{ik}) \| D_{\boldsymbol{\theta}}(\mathbf{x}_i + \mathbf{n}_{ik}, \sigma_{ik}) - \mathbf{x}_i \|_2^2, \text{ where } \{(\sigma_{ik}, \mathbf{n}_{ik})\}_{k=1}^{K} \sim p(\sigma)\mathcal{N}\left(\mathbf{0}, \sigma^2\right). \tag{9}$$

In each iteration of Alg. 1, we are given a (random) set of indices $\mathbb{B}$ of expected size $B$ with no repeated indices, from which we construct a mini-batch $\{\mathbf{x}_i\}_{i\in\mathbb{B}}$. In our implementation (which is based on Yousefpour et al. (2021)) of the Gaussian mechanism for gradient sanitization, we compute the gradient of $l_i$ and apply clipping with norm $C$, and then divide the clipped gradients by the expected batch size $B$ to obtain the batched gradient $G_{batch}$:

$$G_{batch}(\{\mathbf{x}_i\}_{i\in\mathbb{B}}) = \frac{1}{B} \sum_{i\in\mathbb{B}} \texttt{clip}_C\left(\nabla_{\boldsymbol{\theta}} l(\mathbf{x}_i)\right). \tag{10}$$

Finally, Gaussian noise $\mathbf{z} \sim \mathcal{N}(\mathbf{0}, \sigma_{\mathrm{DP}}^2)$ is added to $G_{batch}$ and released as the response $\tilde{G}_{batch}$:

$$\tilde{G}_{batch}(\{\mathbf{x}_i\}_{i\in\mathbb{B}}) = G_{batch}(\{\mathbf{x}_i\}_{i\in\mathbb{B}}) + \frac{C}{B}\mathbf{z}, \quad \mathbf{z} \sim \mathcal{N}(\mathbf{0}, \sigma_{\mathrm{DP}}^2 \boldsymbol{I}) \tag{11}$$

Now, we can restate Theorem 1 as follows with our modified notation:

**Theorem 3.** For noise magnitude $\sigma_{\mathrm{DP}}$, dataset $d = \{\mathbf{x}_i\}_{i=1}^{N}$, and set of (non-repeating) indices $\mathbb{B}$, releasing $\tilde{G}_{batch}(\{\mathbf{x}_i\}_{i\in\mathbb{B}})$ satisfies $\left(\alpha, \alpha/2\sigma_{\mathrm{DP}}^2\right)$-RDP.

*Proof.* Without loss of generality, consider two neighboring datasets $d = \{\mathbf{x}_i\}_{i=1}^N$ and $d' = d \cup \mathbf{x}'$, $\mathbf{x}' \notin d$, and mini-batches $\{\mathbf{x}_i\}_{i \in \mathbb{B}}$ and $\mathbf{x}' \cup \{\mathbf{x}_i\}_{i \in \mathbb{B}}$, where the counter-factual set/batch has one additional entry $\mathbf{x}'$. We can bound the difference of their gradients in $L_2$-norm as:

$$\|G_{batch}(\{\mathbf{x}_i\}_{i \in \mathbb{B}}) - G_{batch}(\mathbf{x}' \cup \{\mathbf{x}_i\}_{i \in \mathbb{B}})\|_2$$

$$= \left\| \frac{1}{B} \sum_{i \in \mathbb{B}} \text{clip}_C (\nabla_{\boldsymbol{\theta}} l(\mathbf{x}_i)) - \left( \frac{1}{B} \text{clip}_C (\nabla_{\boldsymbol{\theta}} l(\mathbf{x}')) + \frac{1}{B} \sum_{i \in \mathbb{B}} \text{clip}_C (\nabla_{\boldsymbol{\theta}} l(\mathbf{x}_i)) \right) \right\|_2$$

$$= \left\| -\frac{1}{B} \text{clip}_C (\nabla_{\boldsymbol{\theta}} l(\mathbf{x}')) \right\|_2$$

$$= \frac{1}{B} \|\text{clip}_C (\nabla_{\boldsymbol{\theta}} l(\mathbf{x}'))\|_2 \leq \frac{C}{B}.$$

We thus have *sensitivity* $S(G_{batch}) = \frac{C}{B}$. Furthermore, since $\mathbf{z} \sim \mathcal{N}(\mathbf{0}, \sigma_{\text{DP}}^2)$, $(C/B)\mathbf{z} \sim \mathcal{N}(\mathbf{0}, (C/B)^2 \sigma_{\text{DP}}^2)$. Following standard arguments, releasing $\tilde{G}_{batch}(\{\mathbf{x}_i\}_{i \in \mathbb{B}}) = G_{batch}(\{\mathbf{x}_i\}_{i \in \mathbb{B}}) + (C/B)\mathbf{z}$ satisfies $(\alpha, \alpha/2\sigma_{\text{DP}}^2)$-RDP (Mironov, 2017). $\square$

In practice, we construct mini-batches by sampling the training dataset for privacy amplification via Poisson Sampling (Mironov et al., 2019), and compute the overall privacy cost of training DPDM via RDP composition (Mironov, 2017). We use these processes as implemented in Opacus (Yousefpour et al., 2021).

For completeness, we also include the Poisson Sampling algorithm in Alg. 2.

---

**Algorithm 2** Poisson Sampling

---

**Input** : Index range $N$, subsampling rate $q$
**Output**: Random batch of indices $\mathbb{B}$ (of expected size $B$)
$\mathbf{c} = \{c_i\}_{i=1}^N \sim \texttt{Bernoulli}(q)$
$\mathbb{B} = \{j : j \in \{1, \dots, N\}, c_j = 1\}$

---

## B  DPGEN ANALYSIS

In this section, we provide a detailed analysis of the privacy guarantees provided in DPGEN (Chen et al., 2022).

As a brief overview, Chen et al. (2022) proposes to learn an energy function $q_\vartheta(\mathbf{x})$ by optimizing the following objective (Chen et al. (2022), Eq. 7):

$$l(\theta; \sigma) = \frac{1}{2} \mathbb{E}_{p(\mathbf{x})} \mathbb{E}_{\tilde{\mathbf{x}} \sim \mathcal{N}(\mathbf{x}, \sigma^2)} \left[ \left\| \frac{\tilde{\mathbf{x}} - \mathbf{x}}{\sigma^2} - \nabla_{\mathbf{x}} \log q_\vartheta(\mathbf{x}) \right\|^2 \right].$$

In practice, the first expectation is replaced by averaging over examples in a private training set $d = \{x_i : x_i \in Y, i \in 1, \dots, m\}$, and $\frac{\tilde{x} - x}{\sigma^2}$ is replaced by $d_i^r = (\tilde{x}_i - x_i^r)/\sigma_i^2$ for each $i$ in $[1, m]$ (not to be confused with $d$ which denotes the dataset in the DP context), where $x_i^r$ is the query response produced by a data-dependent randomized response mechanism.

We believe that there are three errors in DPGEN that renders the privacy gurantee in DPGEN false. We formally prove the first error in the following section, and state the other two errors which are factual but not mathematical. The three errors are:

- The randomized response mechanism employed in DPGEN has a output space that is only supported (has non-zero probability) on combinations of its input *private* dataset. $\epsilon$-differential privacy cannot be achieved as outcomes with non-zero probability[1] can have zero probability when the input dataset is changed by one element. Furthermore, adversaries observing the output can immediately deduce elements of the private dataset.

---

[1] probability over randomness in the privacy mechanism

- The $k$-nearest neighbor filtering used by DPGEN to reduce the number of candidates for the randomized response mechanism is a function of the private data. The likelihood of the $k$-selected set varies with the noisy image $\tilde{x}$ (line 20 of algorithm 1 in DPGEN), and is not correctly accounted for in DPGEN.
- The objective function used to train the denoising network in DPGEN depends on both the ground-truth denoising direction and a noisy image provided to the denoising network. The noisy image is dependent on the training data, and hence leaks privacy. The privacy cost incurred by using this noisy image is not accounted for in DPGEN.

To prove the first error, we begin with re-iterating the formal definition of differential privacy (DP):

**Definition B.1.** ($\epsilon$-Differential Privacy) A randomized mechanism $\mathcal{M} : \mathcal{D} \to \mathcal{I}$ with domain $\mathcal{D}$ and image $\mathcal{I}$ satisfies $(\varepsilon)$-DP if for any two adjacent inputs $d, d' \in \mathcal{D}$ differing by at most one entry, and for any subset of outputs $S \subseteq \mathcal{I}$ it holds that

$$\mathbf{Pr}\left[\mathcal{M}(d) \in S\right] \leq e^{\varepsilon}\mathbf{Pr}\left[\mathcal{M}(d') \in S\right]. \tag{12}$$

The randomized response (RR) mechanism is a fundamental privacy mechanism in differential privacy. A key assumption required in the RR mechanism is that the choices of random response are not dependent on private information, such that when a respondent draws their response randomly from the possible choices, no private information is given. More formally, we give the following definition for randomized response over multiple choices[2]:

**Definition B.2.** Given a fixed response set $Y$ of size $k$. Let $d = \{x_i : x_i \in Y, i \in 1, \ldots, m\}$ be an input dataset. Define "randomized response" mechanism $\mathcal{RR}$ as:

$$\mathcal{RR}(d) = \{G(x_i)\}_{i \in [1,m]} \tag{13}$$

where,

$$G(x_i) = \begin{cases} x_i, \text{ with probability } \frac{e^\epsilon}{e^\epsilon+k-1} \\ x_i' \in Y \setminus x_i, \text{ with probability } \frac{1}{e^\epsilon+k-1} \end{cases}. \tag{14}$$

A classical result is that the mechanism $\mathcal{RR}$ satisfies $\epsilon$-DP (Dwork et al., 2014).

DPGEN considers datasets of the form $d = \{x_i : x_i \in \mathbb{R}^n, i \in 1, \ldots, m\}$. It claims to guarantee differential privacy by applying a stochastic function $H$ to each element of the dataset defined as follows (Eq. 8 of Chen et al. (2022)):

$$\Pr[H(\tilde{x}_i) = w] = \begin{cases} \frac{e^\epsilon}{e^\epsilon+k-1}, \ w = x_i \\ \frac{1}{e^\epsilon+k-1}, \ w = x_i' \in \mathrm{X} \setminus x_i \end{cases},$$

where $\mathrm{X} = \{x_j : max(\tilde{x}_i - x_j)/\sigma_j \leq \beta, x_j \in d\}$ (max is over the dimensions of $\tilde{x}_i - x_j$), $|\mathrm{X}| = k \geq 2$, and $\tilde{x}_i = x_i + z_i, z_i \sim \mathcal{N}(0, \sigma^2 I)$. We first note that $H$ is not only a function of $\tilde{x}_i$ but also $\mathrm{X} \cup x_i$, since its image is determined by $\mathrm{X} \cup x_i$. That is, changes in $X$ will alter the possible outputs of $H$, independently from the value of $\tilde{x}_i$. We make this dependency explicit in our formulation here-forth. This distinction is important as it determines the set of possible outcomes that we need to consider for in the privacy analysis. The authors also noted that $z_i$ is added for training with the denoising objective, not for privacy, so this added Gaussian noise is not essential to the privacy analysis. Furthermore, since $k$ (or equivalently $\beta$) is a hyperparameter that can be tuned, we consider the simpler case where $k = m$, i.e. $\mathrm{X} = d$, as done in the appendix (Eq. 9) by the authors. Thereby we define the privacy mechanism utilized in DPGEN as follows:

**Definition B.3.** Let $d = \{x_i : x_i \in \mathbb{R}^n, i \in 1, \ldots, m\}$ be an input dataset. Define "data dependent randomized response" $\mathcal{M}$ as:

$$\mathcal{M}(d) = \{H(x_i, d)\}_{i \in [1,m]} \tag{15}$$

where,

$$H(x_i, d) = \begin{cases} x_i, \text{ with probability } \frac{e^\epsilon}{e^\epsilon+m-1} \\ x_i' \in d \setminus x_i, \text{ with probability } \frac{1}{e^\epsilon+m-1} \end{cases}. \tag{16}$$

---

[2]This mechanism is analogous to the coin flipping mechanism, where the participant first flip a biased coin to determine whether they'll answer truthfully or lie with probability of lying $\frac{k}{e^\epsilon+k-1}$, and if they were to lie, they then roll a fair $k$ dice to determine the response.

Since the image of $H(x_i, d)$ is $d$, $\mathcal{M}(d)$ is only supported on $d^m$.[3] In other words, the image of $\mathcal{M}$ is data dependent, and any outcome $O$ (which are sets of $\mathbb{R}^n$ tensors, of cardinality $m$) that include elements which are not in $d$ would have a probability of zero to be the outcome of $\mathcal{M}(d)$, i.e. if there exists $z \in O$ and $z \notin d$, then $\Pr[\mathcal{M}(d) = O] = 0$.

To construct our counter-example, we start with considering two neighboring datasets: the training data $d = \{x_i : x_i \in \mathbb{R}^n, i \in 1, \ldots, m\}$, and a counter-factual dataset $d' = \{x_1' : x_1' \in \mathbb{R}^n, x_i : x_i \in \mathbb{R}^n, i \in 2, \ldots, m\}$, differing in their first element ($x_1 \neq x_1'$). Importantly, since differential privacy requires that the likelihoods of outputs to be similar for all valid pairs of neighboring datasets, we are free to assume that elements of $d$ are unique, i.e. no two rows of $d$ are identical.

Another requirement of differential privacy is that the likelihood of any subsets of outputs must be similar, hence we are free to choose any valid response for the counter-example. Thus, letting $O$ denote the outcome of $\mathcal{M}(d)$, we choose $O = d = \{x_1, \ldots, x_m\}$. Clearly, by Definition 0.3, this is a plausible outcome of $\mathcal{M}(d)$ as it is in the support $d^m$. However, $O$ is not in the support of $\mathcal{M}(d')$ since the first element $x_1$ is not in the image of $H(\cdot, d')$; that is $\Pr[H(x, d') = x_1] = 0$ for all $x \in d'$. Privacy protection is violated since any adversary observing $O$ can immediately deduce the participation of $x_1$ in the data release as opposed to any counterfactual data $x_1'$.

More formally, consider response set $T = \{O\} \subset d^m$, and $d^m$ is the image of $\mathcal{M}(d)$, we have

$$\Pr[\mathcal{M}(d) \in T] = \Pr[\mathcal{M}(d) = O] \tag{17}$$

$$= \Pr[H(x_1) = x_1] \prod_{i=2}^{m} \Pr[H(x_i) = x_i] \quad \text{(independent dice rolls)} \tag{18}$$

$$= \frac{e^\epsilon}{e^\epsilon + m - 1} \prod_{i=2}^{m} \Pr[H(x_i) = x_i] \quad \text{(apply definition B.3)} \tag{19}$$

$$> 0 \prod_{i=2}^{m} \Pr[H(x_i) = x_i] \tag{20}$$

$$= \Pr[H(x_1') = x_1] \prod_{i=2}^{m} \Pr[H(x_i) = x_i] \tag{21}$$

$$= \Pr[\mathcal{M}(d') = O] = \Pr[\mathcal{M}(d') \in T]. \tag{22}$$

We can observe that $\Pr[\mathcal{M}(d') \in T] = 0$, as shown in line 9. Clearly, this result violates $\epsilon$-DP for all $\epsilon$, which requires $\Pr[\mathcal{M}(d) \in T] \leq e^\epsilon \Pr[\mathcal{M}(d') \in T]$.

In essence, by using private data to form the response set, we make the image of the privacy mechanism data-dependent. This in turn leaks privacy, since an adversary can immediately rule-out all counter-factual datasets that do not include every element of the response $O$, as these counter-factuals now have likelihood 0. To fix this privacy leak, one could determine a response set a-priori, and use the $\mathcal{RR}$ mechanism in Definition B.2 to privately release data. This modification may not be feasible in practice, since constructing a response set of finite size ($k$) suitable for images is non-trivial. Hence, we believe that it would require fundamental modifications to DPGEN to achieve differential privacy.

Regarding error 2, we point out that in the paragraph following Eq. 8 in DPGEN, $X$ is defined as the set of $k$ points in $d$ that are closest to $\tilde{x}_i$ when weighted by $\sigma_j$. This means that the membership of $X$ is dependent on the value of $\tilde{x}_i$. Thus, any counter-factual input $x_i'$ and $\tilde{x}_i'$ with a different set of $k$ nearest neighbors could have many possible outcomes with 0 likelihood under the true input. In essence, this is a more extreme form of data-dependent randomized response where the response set is dependent on both $d$ and $x_i$.

Regarding error 3, the loss objective in DPGEN (Eq. 7 of DPGEN, $l = \frac{1}{2} E_{p(x)} E_{\tilde{x} \sim N(x, \sigma^2)} \left[ || \frac{\tilde{x} - x}{\sigma^2} - \nabla_x \log q_\theta(\tilde{x}) ||^2 \right]$) includes the term $\nabla_x \log q_\theta(\tilde{x})$, and $\tilde{x}$ is also a function of the private data that is yet to be accounted for at all in the privacy analysis of DPGEN. Hence, one would need to further modify the learning algorithm in DPGEN, such that the inputs to

---

[3]We mean dataset-exponentiation in the sense of repeated cartesian products between sets, i.e. $d^2 = d \otimes d$

Table 7: Four popular DM configs from the literature.

| | VP (Song et al., 2021c) | VE (Song et al., 2021c) | v-prediction (Salimans & Ho, 2022) | Elucidate (Karras et al., 2022) |
|---|---|---|---|---|
| **Network and preconditioning** | | | | |
| Skip scaling $c_{\text{skip}}(\sigma)$ | 1 | 1 | $1/(\sigma^2+1)$ | $\sigma_{\text{data}}^2/(\sigma^2+\sigma_{\text{data}}^2)$ |
| Output scaling $c_{\text{out}}(\sigma)$ | $-\sigma$ | $\sigma$ | $\sigma/\sqrt{1+\sigma^2}$ | $\sigma \cdot \sigma_{\text{data}}/\sqrt{\sigma_{\text{data}}^2+\sigma^2}$ |
| Input scaling $c_{\text{in}}(\sigma)$ | $1/\sqrt{\sigma^2+1}$ | 1 | $1/\sqrt{\sigma^2+1^2}$ | $1/\sqrt{\sigma^2+\sigma_{\text{data}}^2}$ |
| Noise cond. $c_{\text{noise}}(\sigma)$ | $(M-1)t$ | $\ln(\frac{1}{2}\sigma)$ | $t$ | $\frac{1}{4}\ln(\sigma)$ |
| **Training** | | | | |
| Noise distribution | $t \sim \mathcal{U}(\epsilon_t, 1)$ | $\ln(\sigma) \sim \mathcal{U}(\ln(\sigma_{\min}),$ $\ln(\sigma_{\max}))$ | $t \sim \mathcal{U}(\epsilon_{\min}, \epsilon_{\max})$ | $\ln(\sigma) \sim \mathcal{N}(P_{\text{mean}}, P_{\text{std}}^2)$ |
| Loss weighting $\lambda(\sigma)$ | $1/\sigma^2$ | $1/\sigma^2$ | $(\sigma^2+1)/\sigma^2$ ("SNR+1" weighting) | $(\sigma^2+\sigma_{\text{data}}^2)/(\sigma \cdot \sigma_{\text{data}})^2$ |
| **Parameters** | $\beta_d = 19.9, \beta_{\min} = 0.1$ | $\sigma_{\min} = 0.002$ | $\epsilon_{\min} = \frac{2}{\pi}\arccos\frac{1}{\sqrt{1+e^{-13}}}$ | $P_{\text{mean}} = -1.2, P_{\text{std}} = 1.2$ |
| | $\epsilon_t = 10^{-5}, M = 1000$ | $\sigma_{\max} = 80$ | $\epsilon_{\max} = \frac{2}{\pi}\arccos\frac{1}{\sqrt{1+e^{9}}}$ | $\sigma_{\text{data}} = \sqrt{\frac{1}{3}}$ |
| | $\sigma(t) = \sqrt{e^{\frac{1}{2}\beta_d t^2 + \beta_{\min} t} - 1}$ | | $\sigma(t) = \sqrt{\cos^{-2}(\pi t/2) - 1}$ | |

the score model are either processed through an additional privacy mechanism, or sampled randomly without dependence on private data.

Regarding justifying the premise that DPGEN implements the data-dependent randomized response mechanism, we have verified that the privacy mechanism implemented in the repository of DPGEN (https://github.com/chiamuyu/DPGEN[4]) is indeed data-dependent:

In line 30 of losses/dsm.py:

```
sample_ix = random.choices(range(k), weights=weight)[0]
```

randomly selects an index in the range of $[0, k-1]$, which is then used in line 46,

```
sample_buff.append(samples[sample_ix]),
```

to index the private training data and assigned to the output of

```
sample_buff.
```

Values of this variable are then accessed on line 85 to calculate the $\frac{\tilde{x}-x^r}{\sigma^2}$ (as $x^r$) term in the objective function (Chen et al. (2022), Eq. 7).

## C  MODEL AND IMPLEMENTATION DETAILS

### C.1  DIFFUSION MODEL CONFIGS

As discussed in Sec. 2, previous works proposed various denoiser models $D_{\boldsymbol{\theta}}$, noise distributions $p(\sigma)$, and weighting functions $\lambda(\sigma)$. We refer to the triplet $(D_{\boldsymbol{\theta}}, p, \lambda)$ as DM config. In this work, we consider four such configs: *variance preserving* (VP) (Song et al., 2021c), *variance exploding* (VE) (Song et al., 2021c), **v**-prediction (Salimans & Ho, 2022), and the config introduced in Karras et al. (2022) (referred to as *Elucidate* in this work). The triplet for each of these configs can be found in Tab. 7. Note, that we use the parameterization of the denoiser model $D_{\boldsymbol{\theta}}$ from (Karras et al., 2022)

$$D_{\boldsymbol{\theta}}(\mathbf{x}; \sigma) = c_{\text{skip}}(\sigma)\mathbf{x} + c_{\text{out}}(\sigma)F_{\boldsymbol{\theta}}(c_{\text{in}}(\sigma)\mathbf{x}; c_{\text{noise}}(\sigma)), \tag{23}$$

where $F_{\boldsymbol{\theta}}$ is the raw neural network. To accommodate for our particular sampler setting (we require to learn the denoiser model for $\sigma \in [0.002, 80]$; see App. C.3) we slightly modified the parameters of VE and **v**-prediction. For VE, we changed $\sigma_{\min}$ and $\sigma_{\max}$ from 0.02 to 0.002 and from 100 to 80, respectively. For **v**-prediction, we changed $\epsilon_{\min}$ and $\epsilon_{\max}$ from $\frac{2}{\pi}\arccos\frac{1}{\sqrt{1+e^{-20}}}$ to $\frac{2}{\pi}\arccos\frac{1}{\sqrt{1+e^{-13}}}$ and $\frac{2}{\pi}\arccos\frac{1}{\sqrt{1+e^{20}}}$ to $\frac{2}{\pi}\arccos\frac{1}{\sqrt{1+e^{9}}}$, respectively. Furthermore, we cannot base our Elucidate models on the true (training) data standard deviation $\sigma_{\text{data}}$ as releasing this information would result in a privacy cost. Instead, we set $\sigma_{\text{data}}$ to the standard deviation of a uniform distribution between $-1$ and $1$, assuming no prior information on the modeled image data.

---

[4]In particular, we refer to the code at commit: 1f684b9b8898bef010838c6a29c030c07d4a5f87.

Table 8: Model hyperparameters and training details.

| Hyperparameter | MNIST & Fashion-MNIST | CelebA |
|---|---|---|
| **Model** | | |
| Data dimensionality (in pixels) | 28 | 32 |
| Residual blocks per resolution | 2 | 2 |
| Attention resolution(s) | 7 | 8,16 |
| Base channels | 32 | 32 |
| Channel multipliers | 1,2,2 | 1,2,2 |
| EMA rate | 0.999 | 0.999 |
| # of parameters | 1.75M | 1.80M |
| Base architecture | DDPM++ (Song et al., 2021c) | DDPM++ (Song et al., 2021c) |
| **Training** | | |
| # of epochs | 300 | 300 |
| Optimizer | Adam (Kingma & Ba, 2015) | Adam (Kingma & Ba, 2015) |
| Learning rate | $3 \cdot 10^{-4}$ | $3 \cdot 10^{-4}$ |
| Batch size | 4096 | 2048 |
| Dropout | 0 | 0 |
| Clipping constant $C$ | 1 | 1 |
| DP-$\delta$ | $10^{-5}$ | $10^{-6}$ |

### C.1.1 NOISE LEVEL VISUALIZATION

In the following, we provide details on how exactly the noise distributions of the four configs are visualized in Fig. 3. The reason we want to plot these noise distributions is to understand how the different configs assign weight to different noise levels $\sigma$ during training through sampling some $\sigma$'s more and others less. However, to be able to make a meaningful conclusion, we also need to take into account the loss weighting $\lambda(\sigma)$.

Therefore, we consider the effective "importance-weighted" distributions $p(\sigma)\frac{\lambda(\sigma)}{\lambda_{\text{Elucidate}}(\sigma)}$, where we use the loss weighting from the Elucidate config as reference weighting.

The $\frac{\lambda(\sigma)}{\lambda_{\text{Elucidate}}(\sigma)}$ weightings for VP, VE, $\mathbf{v}$-prediction, and Elucidate are then, $\sigma_{\text{data}}^2/(\sigma^2 + \sigma_{\text{data}}^2)$, $\sigma_{\text{data}}^2/(\sigma^2 + \sigma_{\text{data}}^2)$, $\sigma_{\text{data}}^2(\sigma^2 + 1)/(\sigma^2 + \sigma_{\text{data}}^2)$, and 1, respectively. Fig. 3 then visualizes the "importance-weighted" distributions in log-$\sigma$ space, following Karras et al. (2022) (that way, the final visualized log-$\sigma$ distribution of Elucidate remains a normal distribution $\mathcal{N}(P_{\text{mean}}, P_{\text{std}}^2)$).

### C.2 MODEL ARCHITECTURE

We focus on image synthesis and implement the neural network backbone of DPDMs using the DDPM++ architecture (Song et al., 2021c). For class-conditional generation, we add a learned class-embedding to the $\sigma$-embedding as is common practice (Dhariwal & Nichol, 2021). All model hyperparameters and training details can be found in Tab. 8.

### C.3 SAMPLING FROM DIFFUSION MODELS

Let us recall the differential equations we can use to generate samples from DMs:

ODE: $d\mathbf{x} = -\dot{\sigma}(t)\sigma(t)\nabla_{\mathbf{x}} \log p(\mathbf{x}; \sigma(t))\, dt,$        (24)

SDE: $d\mathbf{x} = -\dot{\sigma}(t)\sigma(t)\nabla_{\mathbf{x}} \log p(\mathbf{x}; \sigma(t))\, dt - \beta(t)\sigma^2(t)\nabla_{\mathbf{x}} \log p(\mathbf{x}; \sigma(t))\, dt + \sqrt{2\beta(t)}\sigma(t)\, d\omega_t.$ (25)

Before choosing a numerical sampler, we first need to define a sampling schedule. In this work, we follow Karras et al. (2022) and use the schedule

$$\sigma_i = \left(\sigma_{\max}^{1/\rho} + \frac{i}{M-1}(\sigma_{\min}^{1/\rho} - \sigma_{\max}^{1/\rho})\right)^{\rho}, i \in \{0, \dots, M-1\}, \quad (26)$$

with $\rho=7.0$, $\sigma_{\max}=80$ and $\sigma_{\min}=0.002$. We consider two solvers: the (stochastic/deterministic) DDIM solver (Song et al., 2021a) as well as the stochastic Churn solver introduced in (Karras et al., 2022), for pseudocode see Alg. 3 and Alg. 4, respectively. Both implementations can readily be combined with classifier-free guidance, which is described in App. C.3.1, in which case the denoiser $D_{\boldsymbol{\theta}}(\mathbf{x};\sigma)$ may be replaced by $D_{\boldsymbol{\theta}}^{w}(\mathbf{x};\sigma,\mathbf{y})$, where the guidance scale $w$ is a hyperparameter. Note that the Churn sampler has four additional hyperparameters which should be tuned empirically (Karras et al., 2022). If not stated otherwise, we set $M=1000$ for the Churn sampler and the stochastic DDIM sampler, and $M=50$ for the deterministic DDIM sampler.

---

**Algorithm 3** DDIM sampler (Song et al., 2021a)

---

**Input:** Denoiser $D_{\boldsymbol{\theta}}(\mathbf{x};\sigma)$, Schedule $\{\sigma_i\}_{i\in\{0,\dots,M-1\}}$
**Output:** Sample $\mathbf{x}_M$
Sample $\mathbf{x}_0 \sim \mathcal{N}\left(\mathbf{0},\sigma_0^2\boldsymbol{I}\right)$
**for** $n = 0$ **to** $M-2$ **do**
    Evaluate denoiser $\mathbf{d}_n = D_{\boldsymbol{\theta}}(\mathbf{x}_i,\sigma_i)$
    **if** Solving SDE **then**
        $\mathbf{x}_{n+1} = \mathbf{x}_n + 2\frac{\sigma_{n+1}-\sigma_n}{\sigma_n}(\mathbf{x}_n-\mathbf{d}_n) + \sqrt{2(\sigma_n-\sigma_{n+1})\sigma_n}\mathbf{z}_n, \quad \mathbf{z}_n \sim \mathcal{N}(\mathbf{0},\boldsymbol{I})$
    **else if** Solving ODE **then**
        $\mathbf{x}_{n+1} = \mathbf{x}_n + \frac{\sigma_{n+1}-\sigma_n}{\sigma_n}(\mathbf{x}_n-\mathbf{d}_n)$
    **end if**
**end for**
Return $\mathbf{x}_M = D(\mathbf{x}_{N-1},\sigma_{M-1})$

---

---

**Algorithm 4** Churn sampler (Karras et al., 2022)

---

**Input:** Denoiser $D_{\boldsymbol{\theta}}(\mathbf{x};\sigma)$, Schedule $\{\sigma_i\}_{i\in\{0,\dots,M-1\}}$, $S_{\text{noise}}$, $S_{\text{churn}}$, $S_{\min}$, $S_{\max}$
**Output:** Sample $\mathbf{x}_M$
Set $\sigma_M = 0$
Sample $\mathbf{x}_0 \sim \mathcal{N}\left(\mathbf{0},\sigma_0^2\boldsymbol{I}\right)$
**for** $n = 0$ **to** $M-1$ **do**
    **if** $\sigma_i \in [S_{\min}, S_{\max}]$ **then**
        $\gamma_i = \min(\frac{S_{\text{churn}}}{M}, \sqrt{2}-1)$
    **else**
        $\gamma_i = 0$
    **end if**
    Increase noise level $\widetilde{\sigma}_n = (1+\gamma_n)\sigma_n$
    Sample $\mathbf{z}_n \sim \mathcal{N}\left(\mathbf{0}, S_{\text{noise}}^2\boldsymbol{I}\right)$ and set $\widetilde{\mathbf{x}}_n = \mathbf{x}_n + \sqrt{\widetilde{\sigma}_n^2-\sigma_n^2}\mathbf{z}_n$
    Evaluate denoiser $\mathbf{d}_n = D_{\boldsymbol{\theta}}(\widetilde{\mathbf{x}}_n,\widetilde{\sigma}_n)$ and set $\mathbf{f}_n = \frac{\widetilde{\mathbf{x}}_n-\mathbf{d}_n}{\widetilde{\sigma}_n}$
    $\mathbf{x}_{n+1} = \widetilde{\mathbf{x}}_M + (\sigma_{n+1}-\widetilde{\sigma}_n)\mathbf{f}_n$
    **if** $\sigma_{n+1} \neq 0$ **then**
        Evaluate denoiser $\mathbf{d}'_n = D_{\boldsymbol{\theta}}(\mathbf{x}_{n+1},\sigma_{n+1})$ and set $\mathbf{f}'_n = \frac{\mathbf{x}_{n+1}-\mathbf{d}'_n}{\sigma_{n+1}}$
        Apply second order correction: $\mathbf{x}_{n+1} = \widetilde{\mathbf{x}}_n + \frac{1}{2}(\sigma_{n+1}-\widetilde{\sigma}_n)(\mathbf{f}_n+\mathbf{f}'_n)$
    **end if**
**end for**
Return $\mathbf{x}_M$

---

### C.3.1 GUIDANCE

Classifier guidance (Song et al., 2021c; Dhariwal & Nichol, 2021) is a technique to guide the diffusion sampling process towards a particular conditioning signal $\mathbf{y}$ using gradients, with respect to $\mathbf{x}$, of a pre-trained, noise-conditional classifier $p(\mathbf{y}|\mathbf{x},\sigma)$. Classifier-free guidance (Ho & Salimans, 2021), in contrast, avoids training additional classifiers by mixing denoising predictions of an unconditional and a conditional model, according to a *guidance scale* $w$, by replacing $D_{\boldsymbol{\theta}}(\mathbf{x};\sigma)$ in the score parameterization $s_{\boldsymbol{\theta}} = (D_{\boldsymbol{\theta}}(\mathbf{x};\sigma)-\mathbf{x})/\sigma^2$ with

$$D_{\boldsymbol{\theta}}^{w}(\mathbf{x};\sigma,\mathbf{y}) = (1-w)D_{\boldsymbol{\theta}}(\mathbf{x};\sigma) + wD_{\boldsymbol{\theta}}(\mathbf{x};\sigma,\mathbf{y}). \tag{27}$$

Table 9: DP noise $\sigma_{\mathrm{DP}}$ used for all our experiments.

| $\varepsilon$ | MNIST | Fashion-MNIST | CelebA |
|---|---|---|---|
| 0.2 | 82.5 | 82.5 | N/A |
| 1 | 18.28125 | 18.28125 | 8.82812 |
| 10 | 2.48779 | 2.48779 | 1.30371 |

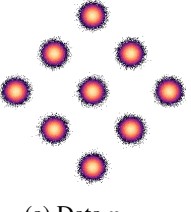 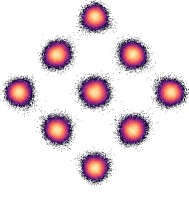 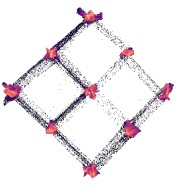

(a) Data $p_{\mathrm{data}}$  (b) Samples from DM.  (c) Samples from GAN.

Figure 6: Mixture of Gaussians: data distribution and (1M) samples from a DM as well as a GAN. Our visualization is based on the log-histogram, which shows single data points as black dots.

$D_{\boldsymbol{\theta}}(\mathbf{x}; \sigma)$ and $D_{\boldsymbol{\theta}}(\mathbf{x}; \sigma, \mathbf{y})$ can be trained jointly; to train $D_{\boldsymbol{\theta}}(\mathbf{x}; \sigma)$ the conditioning signal $\mathbf{y}$ is discarded at random and replaced by a *null token* (Ho & Salimans, 2021). Increased guidance scales $w$ tend to drive samples deeper into the model's modes defined by $\mathbf{y}$ at the cost of sample diversity.

### C.4 HYPERPARAMETERS OF DIFFERENTIALLY PRIVATE DIFFUSION MODELS

Tuning hyperparameters for DP models generally induces a privacy cost which should be accounted for (Papernot & Steinke, 2022). Similar to existing works (De et al., 2022), we neglect the (small) privacy cost associated with hyperparameter tuning. Nonetheless, in this section we want to point out that our hyperparameters show consistent trends across different settings. As a result, we believe our models need little to no hyperparameter tuning in similar settings to the ones considered in this work.

**Model.** We use the DDPM++ (Song et al., 2021c) architecture for all models in this work. Across all three datasets (MNIST, Fashion-MNIST, and CelebA) we found the Elucidate (Karras et al., 2022) DM config to perform best for $\varepsilon = \{1, 10\}$. On MNIST and Fashion-MNIST, we use the **v**-prediction config for $\varepsilon = 0.2$ (not applicable to CelebA).

**DP-SGD training.** In all settings, we use 300 epochs and clipping constant $C = 1$. We use batch size $B = 4096$ for MNIST and Fashion-MNIST and decrease the batch size of CelebA to $B = 2048$ for the sole purpose of fitting the entire batch into GPU memory. The DP noise $\sigma_{\mathrm{DP}}$ values for each setup can be found in Tab. 9

**DM Sampling.** We experiment with different DM solvers in this work. We found the DDIM sampler (Song et al., 2021a) (in particular the stochastic version), which does not have any hyperparameters (without guidance), to perform well across all settings. Using the Churn sampler (Karras et al., 2022), we could improve perceptual quality (measured in FID), however, out of the five (four without guidance) hyperparameters, we only found two (one without guidance) to improve results significantly. We show results for all samplers in App. E.5.

## D TOY EXPERIMENTS

In this section, we describe the details of the toy experiment from paragraph **(ii) Sequential denoising** in Sec. 3.1. For this experiment, we consider a two-dimensional simple Gaussian mixture model of the form

$$p_{\mathrm{data}}(\mathbf{x}) = \sum_{k=1}^{9} \frac{1}{9} p^{(k)}(\mathbf{x}), \tag{28}$$

Table 10: $h$-standard deviation vicinity metric as defined in the paragraph **Fitting** of App. D.

| $h$ | Data | DM | GAN |
|---|---|---|---|
| 1 | 39.4 | 37.2 | 56.8 |
| 2 | 86.5 | 83.3 | 95.3 |
| 3 | 98.9 | 97.7 | 98.9 |
| 4 | 100 | 99.8 | 99.3 |
| 5 | 100 | 100 | 99.6 |
| 6 | 100 | 100 | 99.9 |

where $p^{(k)}(\mathbf{x}) = \mathcal{N}(\mathbf{x}; \boldsymbol{\mu}_k; \sigma_0^2)$ and

$$\boldsymbol{\mu}_1 = \begin{pmatrix} -a \\ 0 \end{pmatrix}, \qquad \boldsymbol{\mu}_2 = \begin{pmatrix} -a/2 \\ a/2 \end{pmatrix}, \quad \boldsymbol{\mu}_3 = \begin{pmatrix} 0 \\ a \end{pmatrix},$$

$$\boldsymbol{\mu}_4 = \begin{pmatrix} -a/2 \\ -a/2 \end{pmatrix}, \quad \boldsymbol{\mu}_5 = \begin{pmatrix} 0 \\ 0 \end{pmatrix}, \qquad \boldsymbol{\mu}_6 = \begin{pmatrix} a/2 \\ a/2 \end{pmatrix},$$

$$\boldsymbol{\mu}_7 = \begin{pmatrix} 0 \\ -a \end{pmatrix}, \qquad \boldsymbol{\mu}_8 = \begin{pmatrix} a/2 \\ -a/2 \end{pmatrix}, \quad \boldsymbol{\mu}_9 = \begin{pmatrix} a \\ 0 \end{pmatrix},$$

where $\sigma_0 = 1/25$ and $a = 1/\sqrt{2}$. The data distribution is visualized in Fig. 6a.

**Fitting.** Initially, we fitted a DM as well as a GAN to the mixture of Gaussians. The neural networks of the DM and the GAN generator use similar ResNet architectures with 267k and 264k (1.1% smaller) parameters, respectively (see App. D.1 for training details). The fitted distributions are visualized in Fig. 6. In this experiment, we use deterministic DDIM (Alg. 3) (Song et al., 2021a), a numerical solver for the Probability Flow ODE (Eq. (1)) (Song et al., 2021c), with 100 neural function evaluations (DDIM-100) as the end-to-end multi-step synthesis process for the DM. Even though our visualization shows that the DM clearly fits the distribution better (Fig. 6), the GAN does not do bad either. Note that our visualization is based on the log-histogram of the sampling distributions, and therefore puts significant emphasis on single data point outliers.

We provide a second method to assess the fitting: In particular, we measure the percentage of points (out of 1M samples) that are within a $h$-standard deviation vicinity of any of the nine modes. A point $\mathbf{x}$ is said to be within a $h$-standard deviation vicinity of the mode $\boldsymbol{\mu}_k$ if $\|\mathbf{x} - \boldsymbol{\mu}_k\| < h\sigma_0$. We present results for this metric in Tab. 10 for $h=\{1, 2, 3, 4, 5, 6\}$. Note that any mode is at least 12.5 standard deviations separated to the next mode, and therefore no point can be in the $h$-standard deviation vicinity of more than two modes for $h \leq 6$.

The results in Tab. 10 indicate that the GAN is slightly too sharp, that is, it puts too many points within the 1- and 2-standard deviation vicinity of modes. Moreover, for larger $h$, the result in Tab. 10 suggests that the samples in Fig. 6c that appear to "connect" the GAN's modes are heavily overemphasized—these samples actually represent less than 1% of the total samples; 99.3% of samples are within a 4-standard deviation vicinity of a mode while modes are at least 12.5 standard deviations separated.

**Complexity.** Now that we have ensured that both the GAN as well as the DM fit the target distribution reasonably well, we can measure the complexity of the DM denoiser $D$, the generator defined by the GAN, as well as the end-to-end multi-step synthesis process (DDIM-100) of the DM. In particular, we measure the complexity of these functions using the Frobenius norm of the Jacobian (Dockhorn et al., 2022b). In particular, we define

$$\mathcal{J}_F(\sigma) = \mathbb{E}_{\mathbf{x} \sim p(\mathbf{x}, \sigma)} \|\nabla_{\mathbf{x}} D_{\boldsymbol{\theta}}(\mathbf{x}, \sigma)\|_F^2. \tag{29}$$

Note that the convolution of a mixture of Gaussian with i.i.d. Gaussian noise is simply the sum of the convolution of the mixture components, i.e.,

$$p(\mathbf{x}; \sigma) = \left(p_{\text{data}} * \mathcal{N}\left(\mathbf{0}, \sigma^2\right)\right)(\mathbf{x}) \tag{30}$$

$$= \sum_{k=1}^{9} \frac{1}{9} \mathcal{N}(\mathbf{x}; \boldsymbol{\mu}_k; \sigma_0^2 + \sigma^2). \tag{31}$$

We then compare $\mathcal{J}_F(\sigma)$ with the complexity of the GAN generator ($S_1$) and the end-to-end synthesis process of the DM ($S_2$). In particular, we define

$$\mathcal{J}_F = \mathbb{E}_{\mathbf{x} \sim \mathcal{N}(\mathbf{0}, \boldsymbol{I})} \|\nabla_{\mathbf{x}} S_i(\mathbf{x})\|_F^2, \quad i \in \{1, 2\}. \tag{32}$$

We want to clarify that for $S_2$ we do not have to backpropagate through an ODE but rather through its discretization, i.e., deterministic DDIM with 100 function evaluations (Alg. 3), since that is how we define the end-to-end multi-step synthesis process of the DM in this experiment. Furthermore, we chose the latent space of the GAN to be two-dimensional such that $\nabla_{\mathbf{x}} S_i(\mathbf{x}) \in \mathbb{R}^{2 \times 2}$ for both the GAN and the DM; this ensures a fair comparison. The final complexities are visualized in Fig. 2.

### D.1 TRAINING DETAILS

**DM training.** Training the diffusion model is very simple. We use the Elucidate config and train for 50k iterations (with batch size $B{=}256$) using Adam with learning rate $3 \cdot 10^{-4}$. We use an EMA rate of 0.999.

**GAN training.** Training GANs on two-dimensional mixture of Gaussians is notoriously difficult (see, for example, Sec. 5.1 in (Yazıcı et al., 2019)). We experimented with several setups and found the following to perform well: We train for 50k iterations (with batch size $B{=}256$) using Adam with learning rate $3 \cdot 10^{-4}$ and ($\beta_1{=}0.0$, $\beta_2 = 0.9$) for both the generator and the discriminator. Following Yazıcı et al. (2019), we use EMA (rate of 0.999 as in the DM). We found it crucial to make the discriminator bigger than the generator; in particular, we use twice as many hidden layers in the discriminator's ResNet. Furthermore, we use `ReLU` and `LeakyReLU` for the generator and the discriminator, respectively.

## E  IMAGE EXPERIMENTS

### E.1 EVALUATION METRICS, BASELINES, AND DATASETS

**Metrics.** We measure sample quality via Fréchet Inception Distance (FID) (Heusel et al., 2017). We follow the DP generation literature and use 60k generated samples. The particular Inception-v3 model used for FID computation is taken from Karras et al. (2021)[5]. On MNIST and Fashion-MNIST, we follow the standard procedure of repeating the channel dimension three times before feeding images into the Inception-v3 model.

On MNIST and Fashion-MNIST, we additionally assess the utility of generated data by training classifiers on synthesized samples and compute class prediction accuracy on real data. Similar to previous works, we consider three classifiers: logistic regression (Log Reg), MLP, and CNN classifiers. The model architectures are taken from the `DP-Sinkhorn` repository (Cao et al., 2021).

For downstream classifier training, we follow the DP generation literature and use 60k synthesized samples. We follow Cao et al. (2021) and split the 60k samples into a training set (90%) and a validation set (remaining 10%). We train all models for 50 epochs, using Adam with learning rate $3 \cdot 10^{-4}$. We regularly save checkpoints during training and use the checkpoint that achieves the best accuracy on the validation split for final evaluation. Final evaluation is performed on real, non-synthetic data. We train all models for 50 epochs, using Adam with learning rate $3 \cdot 10^{-4}$.

**Baselines.** We run baseline experiments for PEARL (Liew et al., 2022). In particular, we train models for $\varepsilon{=}\{0.2, 1, 10\}$ on MNIST and Fashion-MNIST. We confirmed that our models match the performance reported in their paper. In fact, our models perform slightly better (in terms of the LeNet-FID metric Liew et al. (2022) uses). We then follow the same evaluation setup (see **Metrics** above) as for our DPDMs. Most importantly, we use the standard Inception network-based FID calculation, similarly as most works in the (DP) image generative modeling literature.

**Datasets.** We use three datasets in this work: MNIST (LeCun et al., 2010), Fashion-MNIST (Xiao et al., 2017) and CelebA (Liu et al., 2015).

---

[5]https://api.ngc.nvidia.com/v2/models/nvidia/research/stylegan3/versions/1/files/metrics/inception-2015-12-05.pkl

Table 11: Noise multiplicity ablation on MNIST and Fashion-MNIST.

| $K$ | MNIST | | | | Fashion-MNIST | | | |
| | FID | Acc (%) | | | FID | Acc (%) | | |
| | | Log Reg | MLP | CNN | | Log Reg | MLP | CNN |
|---|---|---|---|---|---|---|---|---|
| 1 | 76.9 | 84.2 | 87.5 | 91.7 | 72.5 | 76.0 | 76.3 | 75.9 |
| 2 | 60.1 | 84.8 | 88.3 | 93.1 | 61.4 | 76.7 | 77.0 | 77.4 |
| 4 | 57.1 | 85.2 | 88.0 | 92.8 | 61.1 | 76.7 | 77.2 | 77.0 |
| 8 | 44.8 | 86.2 | 89.2 | 94.1 | 58.2 | 75.2 | 76.3 | 77.4 |
| 16 | 36.9 | 86.0 | 89.8 | 94.2 | 58.5 | 77.0 | 77.4 | 78.8 |
| 32 | 34.8 | 86.8 | 90.1 | 94.4 | 57.7 | 76.4 | 77.0 | 77.1 |

### E.2 COMPUTATIONAL RESOURCES

For all experiments, we use an in-house GPU cluster of V100 NVIDIA GPUs. On eight GPUs, models on MNIST and Fashion-MNIST trained for roughly one day and models on CelebA for roughly four days. We tried to maximize performance by using a large number of epochs, which results in a good privacy-utility trade-off, as well as high noise multiplicity; this results in relatively high training time (when compared to existing DP generative models).

Models with very little drop in downstream accuracy can be trained in much less time by decreasing the noise multiplicity: for example, on MNIST for $\varepsilon{=}1$, the CNN-classifier accuracy only drops by 2.7% (from 94.4% to 91.7%) when decreasing $K = 32$ to $K = 1$ (32-fold speed-up); see Tab. 11. On the other hand, the FID metric suffers considerably when decreasing the noise multiplicity.

### E.3 TRAINING DP-SGD CLASSIFIERS

We train classifiers on MNIST and Fashion-MNIST using DP-SGD directly. We follow the setup used for training DPDMs, in particular, batchsize $B = 4096$, 300 epochs and clipping constant $C = 1$. Recently, De et al. (2022) found EMA to be helpful in training image classifiers: we follow this suggestion and use an EMA rate of 0.999 (same rate as used for training DPDMs).

### E.4 EXTENDED QUANTITATIVE RESULTS

In this section, we show additional quantitative results not presented in the main paper. In particular, we present extended results for all ablation experiments.

#### E.4.1 NOISE MULTIPLICITY

In the main paper, we present noise multiplicity ablation results on MNIST with $\varepsilon{=}1$ (Tab. 3). All results for MNIST and Fashion-MNIST on all three privacy settings ($\varepsilon{=}\{0.2, 1, 10\}$) can be found in Tab. 11.

#### E.4.2 DIFFUSION MODEL CONFIG

In the main paper, we present DM config ablation results on MNIST with $\varepsilon{=}0.2$ (Tab. 3). All results for MNIST and Fashion-MNIST on all three privacy settings ($\varepsilon{=}\{0.2, 1, 10\}$) can be found in Tab. 12.

#### E.4.3 DIFFUSION SAMPLER GRID SEARCH AND ABLATION

**Churn sampler grid search.** We run a small grid search for the hyperparameters of the Churn sampler (together with the guidance weight $w$ for classifier-free guidance). For MNIST and Fashion-MNIST on $\varepsilon{=}0.2$ we run a two-stage grid search. Using $S_{\min}{=}0.05$, $S_{\max}{=}50$, and $S_{\text{noise}}{=}1$, which we found to be sensible starting values, we ran an initial grid search over $w{=}\{0, 0.125, 0.25, 0.5, 1.0, 2.0\}$ and $S_{\text{churn}}{=}\{0, 5, 10, 25, 50, 100, 150, 200\}$, which we found to be the two most critical hyperparameters of the Churn sampler. Afterwards, we ran a second

Table 12: DM config ablation.

| Method | DP-$\varepsilon$ | MNIST | | | | Fashion-MNIST | | | |
|---|---|---|---|---|---|---|---|---|---|
| | | FID | Acc (%) | | | FID | Acc (%) | | |
| | | | Log Reg | MLP | CNN | | Log Reg | MLP | CNN |
| VP (Song et al., 2021c) | 0.2 | 197 | 23.1 | 25.5 | 24.2 | 146 | 49.7 | 51.6 | 51.7 |
| VE (Song et al., 2021c) | 0.2 | 171 | 17.9 | 15.4 | 13.9 | 178 | 22.2 | 27.9 | 49.4 |
| V-prediction (Salimans & Ho, 2022) | 0.2 | 97.8 | 80.2 | 81.3 | 84.4 | 115 | 71.3 | 70.9 | 71.8 |
| Elucidate (Karras et al., 2022) | 0.2 | 119 | 62.4 | 67.3 | 49.2 | 93.5 | 64.7 | 65.9 | 66.6 |
| VP (Song et al., 2021c) | 1 | 82.2 | 59.4 | 69.3 | 72.6 | 73.4 | 68.3 | 70.4 | 72.7 |
| VE (Song et al., 2021c) | 1 | 165 | 17.9 | 20.5 | 26.0 | 156 | 30.7 | 36.0 | 49.8 |
| V-prediction (Salimans & Ho, 2022) | 1 | 34.8 | 86.8 | 90.1 | 94.4 | 57.7 | 76.4 | 77.0 | 77.1 |
| Elucidate (Karras et al., 2022) | 1 | 34.2 | 86.2 | 90.1 | 94.9 | 47.1 | 77.4 | 78.0 | 79.4 |
| VP (Song et al., 2021c) | 10 | 12.3 | 88.8 | 94.1 | 97.0 | 22.3 | 81.2 | 81.6 | 84.5 |
| VE (Song et al., 2021c) | 10 | 88.6 | 48.0 | 56.9 | 63.8 | 83.2 | 69.0 | 70.4 | 75.4 |
| V-prediction (Salimans & Ho, 2022) | 10 | 7.65 | 90.4 | 94.4 | 97.7 | 23.1 | 82.0 | 83.7 | 85.5 |
| Elucidate (Karras et al., 2022) | 10 | 6.13 | 90.4 | 94.6 | 97.5 | 17.4 | 82.6 | 84.1 | 86.2 |

Table 13: Diffusion sampler comparison. We compare the Churn sampler (Karras et al., 2022) to stochastic and determistic DDIM (Song et al., 2021a).

| Sampler | DP-$\varepsilon$ | MNIST | | | | Fashion-MNIST | | | |
|---|---|---|---|---|---|---|---|---|---|
| | | FID | Acc (%) | | | FID | Acc (%) | | |
| | | | Log Reg | MLP | CNN | | Log Reg | MLP | CNN |
| Churn (FID) | 0.2 | **61.9** | 65.3 | 65.8 | 71.9 | **78.4** | 53.6 | 55.3 | 57.0 |
| Churn (Acc) | 0.2 | 104 | 81.0 | 81.7 | **86.3** | 128 | 70.4 | 71.3 | **72.3** |
| Stochastic DDIM | 0.2 | 97.8 | 80.2 | 81.3 | 84.4 | 115 | 71.3 | 70.9 | 71.8 |
| Deterministic DDIM | 0.2 | 120 | **81.3** | **82.1** | 84.8 | 132 | **71.5** | **71.6** | 71.8 |
| Churn (FID) | 1 | **23.4** | 83.8 | 87.0 | 93.4 | **37.8** | 71.5 | 71.7 | 73.6 |
| Churn (Acc) | 1 | 35.5 | **86.7** | 91.6 | **95.3** | 51.4 | 76.3 | 76.9 | **79.4** |
| Stochastic DDIM | 1 | 34.2 | 86.2 | 90.1 | 94.9 | 47.1 | 77.4 | 78.0 | **79.4** |
| Deterministic DDIM | 1 | 50.4 | 85.7 | **91.8** | 94.9 | 60.6 | **77.5** | **78.2** | 78.9 |
| Churn (FID) | 10 | **5.01** | 90.5 | 94.6 | 97.3 | 18.6 | 80.4 | 81.1 | 84.9 |
| Churn (Acc) | 10 | 6.65 | **90.8** | 94.8 | **98.1** | 19.1 | 81.1 | 83.0 | **86.2** |
| Stochastic DDIM | 10 | 6.13 | 90.4 | 94.6 | 97.5 | **17.4** | **82.6** | **84.1** | **86.2** |
| Deterministic DDIM | 10 | 10.9 | 90.5 | **95.2** | 97.7 | 19.7 | 81.9 | 83.9 | **86.2** |

grid search over $S_{\text{noise}}=\{1, 1.005\}$, $S_{\text{min}}=\{0.01, 0.02, 0.05, 0.1, 0.2\}$, and $S_{\text{max}}=\{10, 50, 80\}$ using the best $(w, S_{\text{churn}})$ setting for each of the two models. For MNIST and Fashion-MNIST on $\varepsilon=\{1, 10\}$, we ran a single full grid search over $w=\{0, 0.25, 0.5, 1.0, 2.0\}$, $S_{\text{churn}}=\{10, 25, 50, 100\}$, and $S_{\text{min}}=\{0.025, 0.05, 0.1, 0.2\}$ while setting $S_{\text{noise}}=1$. For CelebA, on both $\varepsilon=1$ and $\varepsilon=10$, we also ran a single full grid search over $S_{\text{churn}}=\{50, 100, 150, 200\}$, and $S_{\text{min}}=\{0.005, 0.05\}$ while setting $S_{\text{noise}}=1$. The best settings for FID metric and downstream CNN accuracy can be found in Tab. 14 and Tab. 15, respectively.

Throughout all experiments we found two consistent trends that are listed in the following:

- If optimizing for FID, set $S_{\text{churn}}$ relatively high and $S_{\text{min}}$ relatively small. Increase $S_{\text{churn}}$ and decrease $S_{\text{min}}$ as $\varepsilon$ is decreased.

- If optimizing for downstream accuracy, set $S_{\text{churn}}$ relatively small and $S_{\text{min}}$ relatively high.

**Sampling ablation.** In the main paper, we present a sampler ablation for MNIST (Tab. 6). Results for Fashion-MNIST (as well as) MNIST can be found in Tab. 13.

### E.5 EXTENDED QUALITATIVE RESULTS

In this section, we show additional generated samples by our DPDMs. On MNIST, see Fig. 7, Fig. 8, and Fig. 9 for $\varepsilon=10$, $\varepsilon=1$, and $\varepsilon=0.2$, respectively. On Fashion-MNIST, see Fig. 10, Fig. 11, and

Table 14: Best Churn sampler settings for FID metric.

| Parameter | MNIST | | | Fashion-MNIST | | | CelebA | |
|---|---|---|---|---|---|---|---|---|
| | $\varepsilon{=}0.2$ | $\varepsilon{=}1$ | $\varepsilon{=}10$ | $\varepsilon{=}0.2$ | $\varepsilon{=}1$ | $\varepsilon{=}10$ | $\varepsilon{=}1$ | $\varepsilon{=}10$ |
| $w$ | 1 | 0 | 0.25 | 2 | 1 | 0.25 | N/A | N/A |
| $S_{\text{churn}}$ | 200 | 100 | 50 | 150 | 50 | 25 | 200 | 50 |
| $S_{\min}$ | 0.01 | 0.05 | 0.05 | 0.02 | 0.025 | 0.2 | 0.005 | 0.005 |
| $S_{\max}$ | 50 | 50 | 50 | 10 | 50 | 50 | 50 | 50 |
| $S_{\text{noise}}$ | 1 | 1 | 1 | 1 | 1 | 1 | 1 | 1 |

Table 15: Best Churn sampler settings for downstream CNN accuracy.

| Parameter | MNIST | | | Fashion-MNIST | | |
|---|---|---|---|---|---|---|
| | $\varepsilon{=}0.2$ | $\varepsilon{=}1$ | $\varepsilon{=}10$ | $\varepsilon{=}0.2$ | $\varepsilon{=}1$ | $\varepsilon{=}10$ |
| $w$ | 0.125 | 0 | 0 | 0.125 | 0 | 0 |
| $S_{\text{churn}}$ | 10 | 10 | 10 | 5 | 10 | 10 |
| $S_{\min}$ | 0.2 | 0.1 | 0.025 | 0.02 | 0.025 | 0.1 |
| $S_{\max}$ | 10 | 50 | 50 | 80 | 50 | 50 |
| $S_{\text{noise}}$ | 1.005 | 1 | 1 | 1.005 | 1 | 1 |

Fig. 12 for $\varepsilon{=}10$, $\varepsilon{=}1$, and $\varepsilon{=}0.2$, respectively. On CelebA, see Fig. 13 and Fig. 14 for $\varepsilon{=}10$ and $\varepsilon{=}1$, respectively.

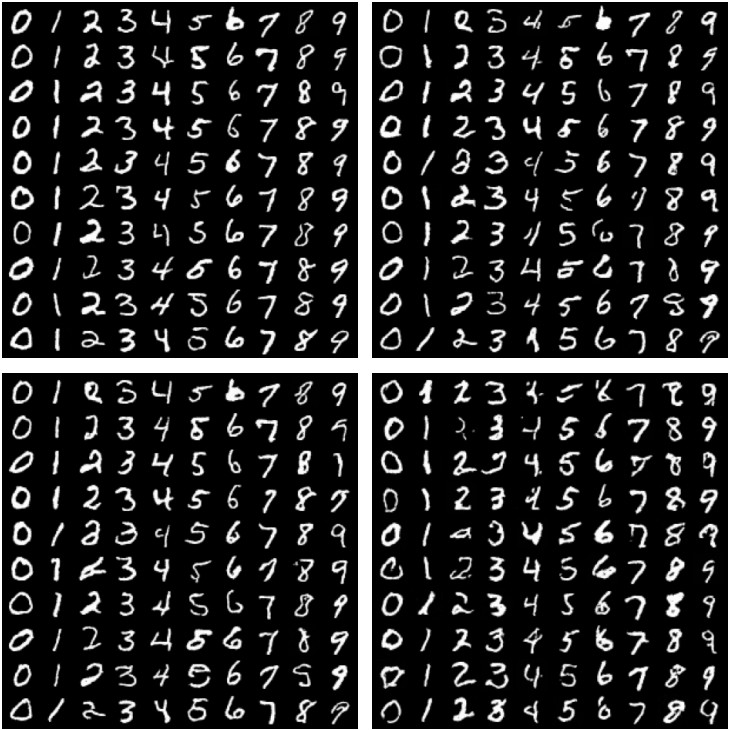

Figure 7: Additional images generated by DPDM on MNIST for $\varepsilon=10$ using Churn (FID) (*top left*), Churn (Acc) (*top right*), stochastic DDIM (*bottom left*), and deterministic DDIM (*bottom right*).

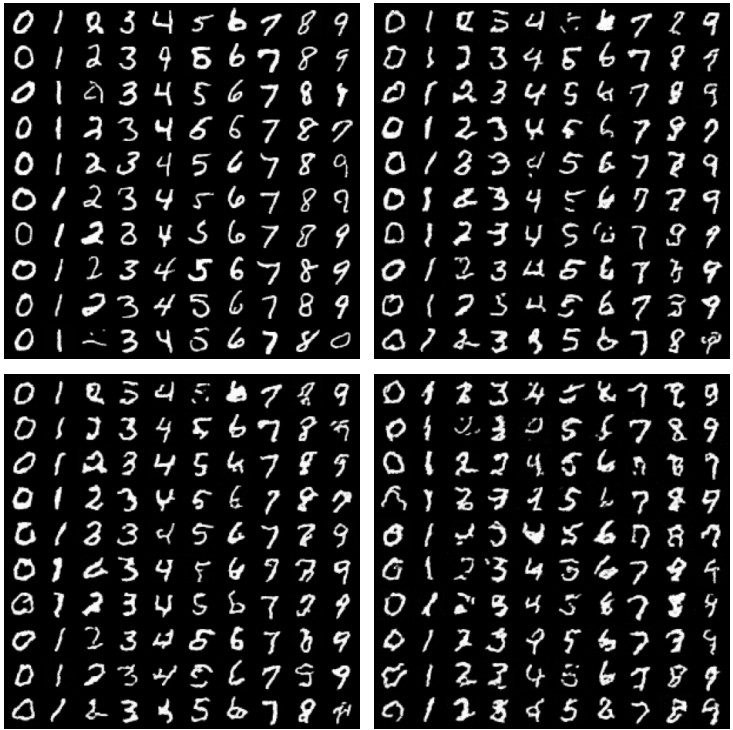

Figure 8: Additional images generated by DPDM on MNIST for $\varepsilon=1$ using Churn (FID) (*top left*), Churn (Acc) (*top right*), stochastic DDIM (*bottom left*), and deterministic DDIM (*bottom right*).

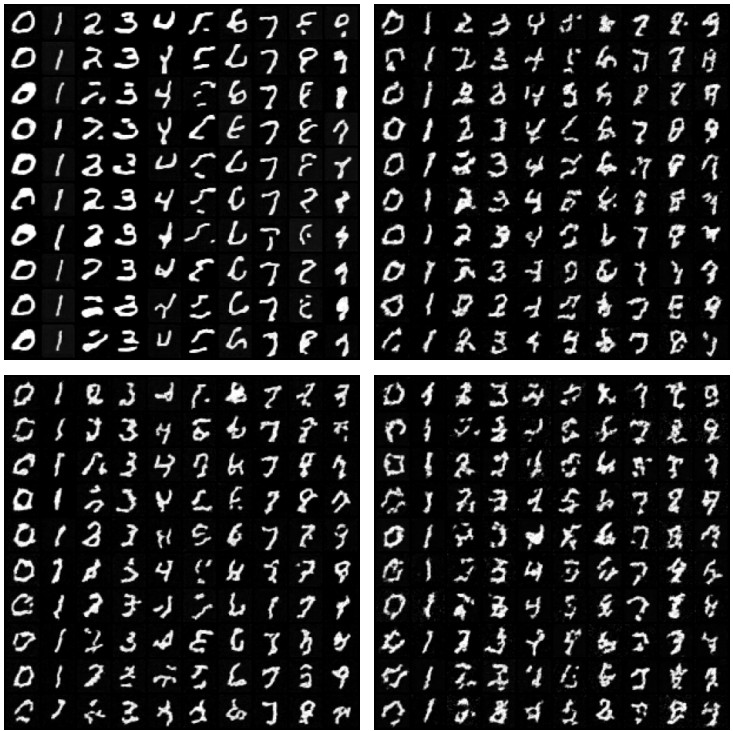

Figure 9: Additional images generated by DPDM on MNIST for $\varepsilon=0.2$ using Churn (FID) (*top left*), Churn (Acc) (*top right*), stochastic DDIM (*bottom left*), and deterministic DDIM (*bottom right*).

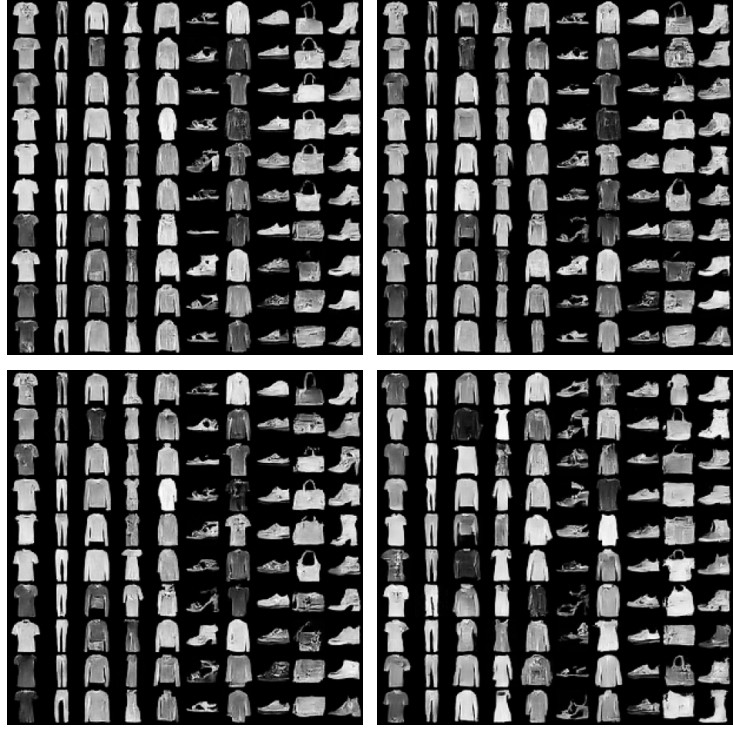

Figure 10: Additional images generated by DPDM on Fashion-MNIST for $\varepsilon=10$ using Churn (FID) (*top left*), Churn (Acc) (*top right*), stochastic DDIM (*bottom left*), and deterministic DDIM (*bottom right*).

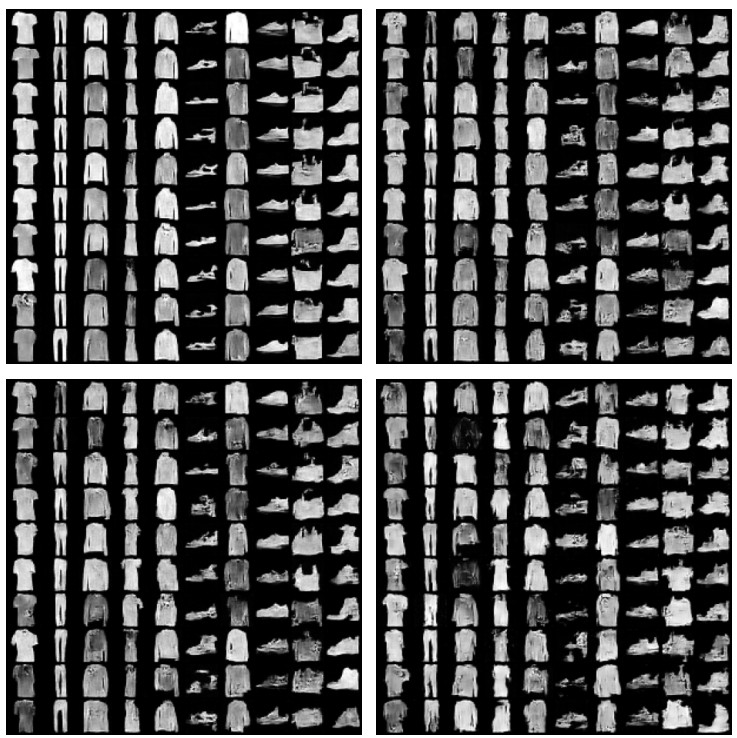

Figure 11: Additional images generated by DPDM on Fashion-MNIST for $\varepsilon=1$ using Churn (FID) (*top left*), Churn (Acc) (*top right*), stochastic DDIM (*bottom left*), and deterministic DDIM (*bottom right*).

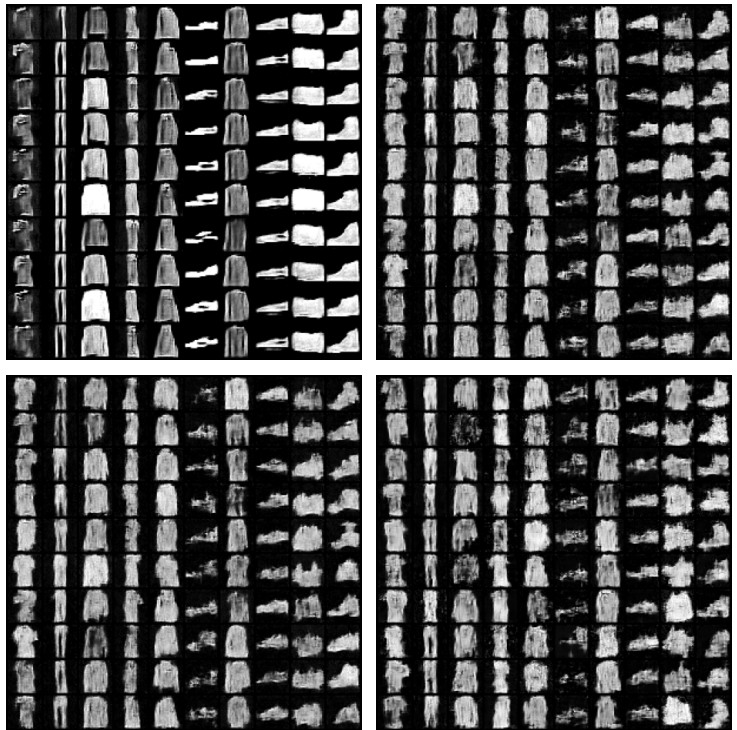

Figure 12: Additional images generated by DPDM on Fashion-MNIST for $\varepsilon=0.2$ using Churn (FID) (*top left*), Churn (Acc) (*top right*), stochastic DDIM (*bottom left*), and deterministic DDIM (*bottom right*).

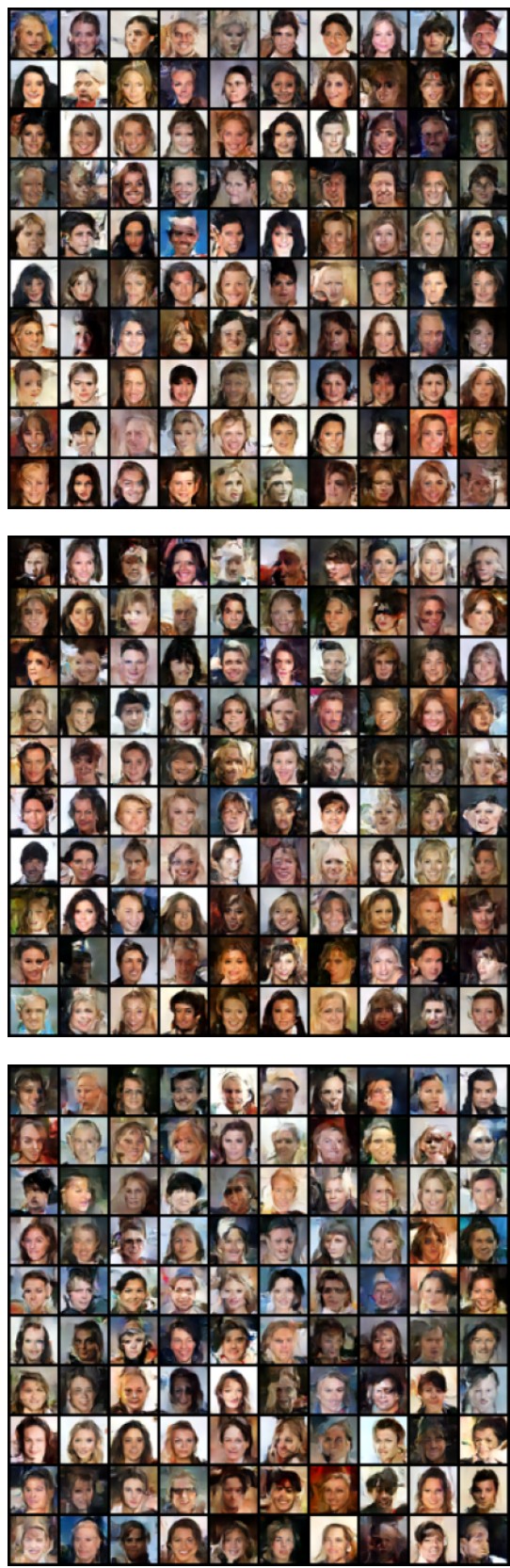

Figure 13: Additional images generated by DPDM on CelebA for $\varepsilon{=}10$ using Churn (*top*), stochastic DDIM (*middle*), and deterministic DDIM (*bottom*).

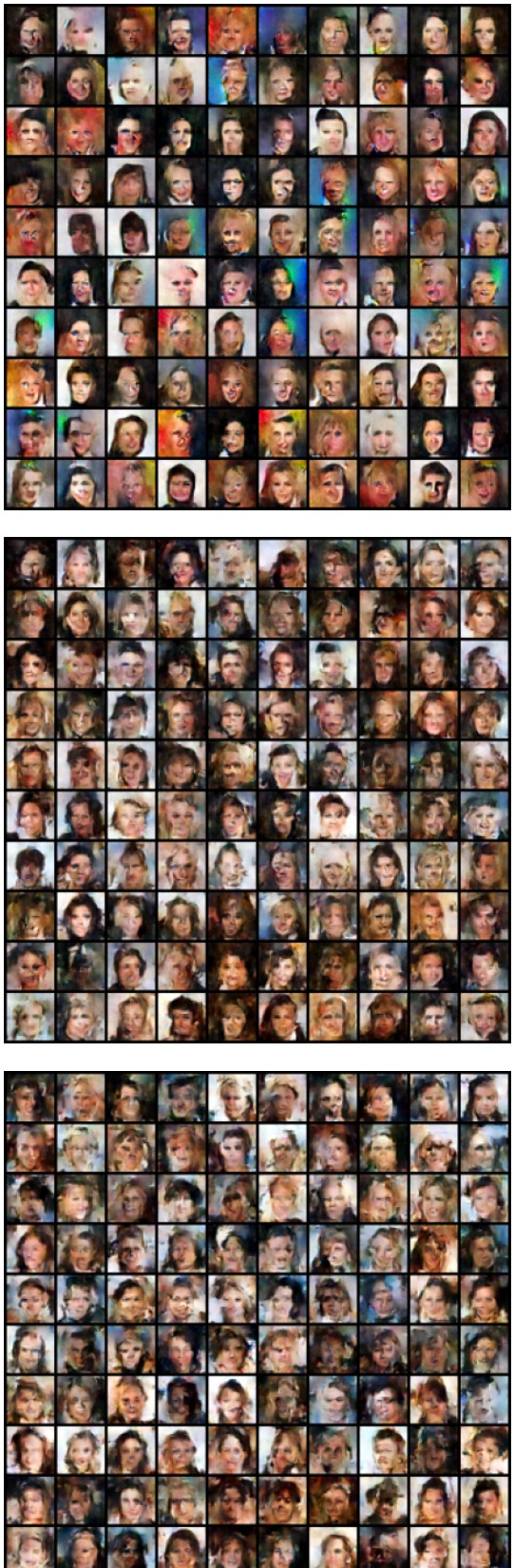

Figure 14: Additional images generated by DPDM on CelebA for $\varepsilon=1$ using Churn (*top*), stochastic DDIM (*middle*), and deterministic DDIM (*bottom*).

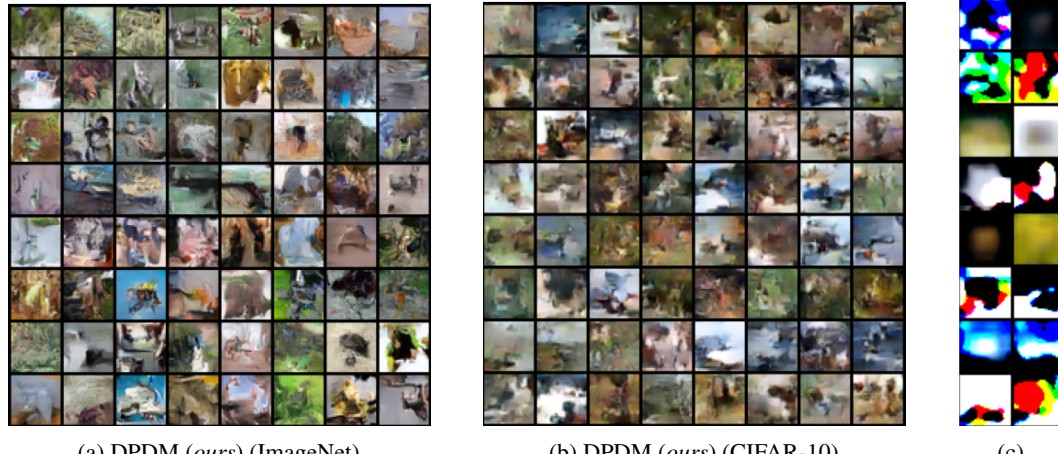

(a) DPDM (*ours*) (ImageNet)   (b) DPDM (*ours*) (CIFAR-10)   (c)

Figure 15: Additional experiments on challenging diverse datasets. Samples from our DPDM on ImageNet and CIFAR-10, as well as CIFAR-10 samples from DP-MERF (Harder et al., 2021) in (c).

## F    REBUTTAL DISCUSSIONS

In this section, we provide additional content related to questions and concerns raised by the reviewers. If the paper will be accepted, we will re-organize the content and integrate some of the discussions here into the main text.

### F.1    ADDITIONAL EXPERIMENTS ON DIVERSE DATASETS

We provide results for additional experiments on challenging diverse datasets, namely, CIFAR-10 (Krizhevsky, 2009) and ImageNet (Deng et al., 2009) (resolution 32x32), both in the class-conditional setting similar to our other experiments on MNIST and Fashion-MNIST. To the best of our knowledge, we are the first to attempt pure DP image generation on ImagenNet.

For both experiments, we use the same neural network architecture as for CelebA (32x32) in the main paper; see model hyperparameters in Tab. 8. On CIFAR-10, we train for 500 epochs using noise multiplicity $K = 32$ under the privacy setting ($\varepsilon = 10, \delta = 10^{-5}$). In ImageNet, we train for 100 epochs using noise multiplicity $K = 8$ under the privacy setting ($\varepsilon = 10, \delta = 7 \cdot 10^{-7}$); given the limited time, training for longer (or using larger $K$) was not possible on ImageNet due to its sheer size. We achieve FIDs of 97.7 and 61.3 for CIFAR-10 and ImageNet, respectively. No previous works reported FID scores on these datasets and for these privacy settings, but we hope that our scores can serve as reference points for future work. In Fig. 15, we show samples for both datasets from our DPDMs and visually compare to an existing DP generative modeling work on CIFAR-10, DP-MERF (Harder et al., 2021). Our DPDMs cannot learn clear objects; however, overall image/pixel statistics seem to be captured correctly. In contrast, the DP-MERF baseline collapses entirely. We are not aware of any other works tackling these tasks. Hence, we believe that DPDMs represent a major step forward.

### F.2    ADDITIONAL EXPERIMENTS AT HIGHER RESOLUTION

We provide results for additional experiments on CelebA at higher resolution (64x64). To accommodate the higher resolution, we added an additional upsampling/downsampling layer to the U-Net, which results in roughly a 11% increase in the number of parameters, from 1.80M to 2.00M parameters. The only row that changed in the CelebA model hyperparameter table (Tab. 8) is the one about the channel multipliers. It is adapted from (1,2,2) to (1,2,2,2). We train for 300 epochs using $K = 8$ under the privacy setting ($\varepsilon = 10, \delta = 10^{-6}$). We achieve an FID of 78.3 (again, for reference; no previous works reported quantitative results on this task). In Fig. 16, we show samples and visually compare to existing DP generative modeling work on CelebA at 64x64 resolution. Although the faces generated by our DPDM are somewhat distorted, the model overall is able to clearly generate

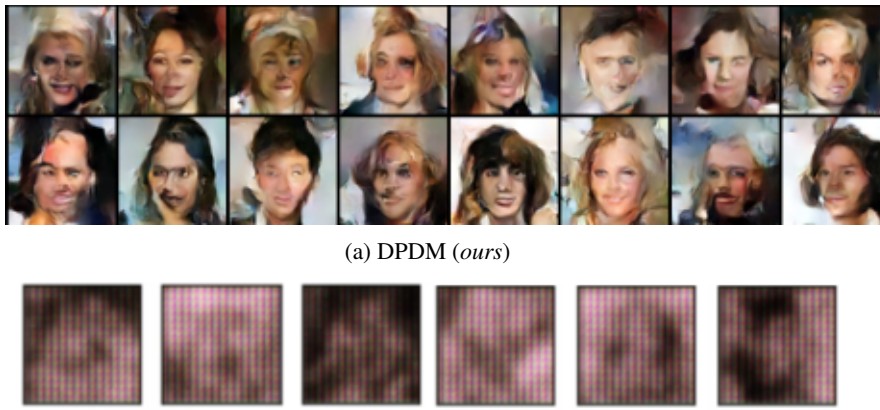

(a) DPDM (*ours*)

(b) DataLens (Wang et al., 2021)

Figure 16: Additional experiments on CelebA at higher resolution (64x64). Samples from our method and DataLens (Wang et al., 2021).

face-like structures. In contrast, DataLens generates incoherent very low quality outputs. No other existing works tried generating 64x64 CelebA images with rigorous DP guarantees, to the best of our knowledge. Also this experiments implies that DPDMs can be considered a major step forward for DP generative modeling.

### F.3 CONCERNS REGARDING THE CELEBA BENCHMARK

As correctly pointed out by one of the reviewers, CelebA contains multiple images per person, whereas our method considers the per-image privacy guarantee. For an individual with $k$ images in the dataset, a model with $(\varepsilon, \delta)$ per-image DP affords $(k\varepsilon, ke^{(k-1)\varepsilon}\delta)$-DP to the individual according to the Group Privacy theorem (Dwork et al., 2014). We leave a more rigorous study of DPDMs with Group Privacy to future research and note that CelebA serves as a standard benchmark in our work.

### F.4 VARIANCE REDUCTION VIA NOISE MULTIPLICITY

As discussed in Sec. 3.2, we introduce *noise multiplicity* to reduce gradient variance. In Fig. 17, we plot the (estimated) variance of the Monte Carlo estimator that is obtained after applying the parameter gradient operation on Eq. (7) for all model parameter gradients in a histogram. Specifically, for each $K$ we plot one histogram and each histogram corresponds to the variances for all the different model parameter gradients. We use our trained model on MNIST and set $\mathbf{x}$ to a randomly sampled MNIST image. We can clearly see that increasing $K$ leads to variance reduction. We estimate the variance of the gradient estimators (that use K samples) using 1000 Monte Carlo estimates.

Note that a variance reduction effect is also expected by theory: When calculating gradients of our training objective, we are effectively replacing expectations with Monte Carlo estimates, as is common practice to ensure numerical tractability. For a generic function $r$ over distribution $p(\mathbf{k})$, we have $\mathbb{E}_{p(\mathbf{k})}[r(\mathbf{k})] \approx \frac{1}{m}\sum_{i=1}^{m} r(\mathbf{k}_i)$, where $\{\mathbf{k}_i\}_{i=1}^{m} \sim p(\mathbf{k})$ (Monte Carlo estimator for expectation of function $r$ with respect to distribution $p$). The Monte Carlo estimate is a noisy unbiased estimator of the expectation $\mathbb{E}_{p(\mathbf{k})}[r(\mathbf{k})]$ with variance $\frac{1}{m}\mathrm{Var}_p[r]$, where $\mathrm{Var}_p[r]$ is the variance of $r$ itself. This is a well-known fact; see for example Chapter 2 of the excellent book by Owen (2013). In our noise multiplicity, $K$ acts like $m$ here and correspondingly reduces the variance or the estimator (in our noise multiplicity case, the parameter gradient estimator).

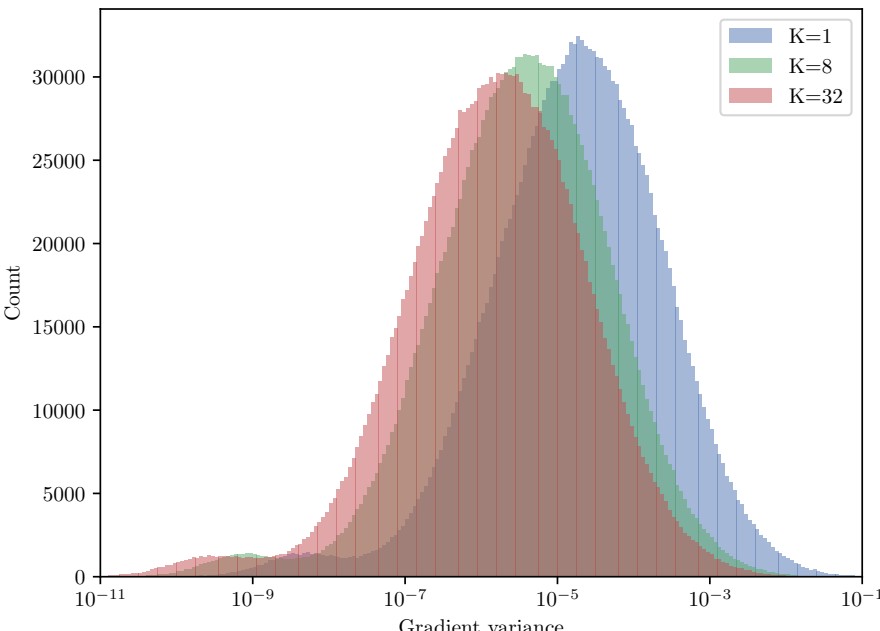

Figure 17: Histogram of gradient variance over all parameters of the model. Increasing $K$ in *noise multiplicity* clearly leads to variance reduction. Note the logarithmic x-axis (the variance reduction is significant).

