# OpenReview forum: "Differentially Private Diffusion Models"
_ICLR.cc/2023/Conference — Submitted to ICLR 2023_

### Official Review · Reviewer_vz3p · 2022-10-17

**Confidence:** 4
**Clarity, Quality, Novelty And Reproducibility:** Please see Strength And Weaknesses.
**Correctness:** 3
**Technical Novelty And Significance:** 3
**Empirical Novelty And Significance:** 4
**Recommendation:** 8

**Strength And Weaknesses:**

**Strength**

1. This is the first work to use diffusion models for DP data generation. The authors achieve state-of-the-art performance on several image benchmarks.

2. The authors propose several novel modifications to the non-private implementation of diffusion models, which are necessary to achieve good performance.

3. The paper is well written. I enjoy reading it.

**Weaknesses**

I have one concern about the benchmarks used in this paper. They have a common pattern: every image has very visually similar neighbors. For MNIST, all digits from the same class look very similar. For CelebA, there are dozens of images for most of the identities. I understand these two datasets are standard in the literature, but I think they are not representative enough. My suggestion is to include some other datasets that do not have clear clusters in the raw input space, e.g., CIFAR-10 which is also used in some previous works. I don't expect the results to be as good as those in the current version, but it is good for the community to know the results.

Another small concern about CelebA is that the privacy guarantee is example-wise. Each identity has dozens of examples so the identity-wise privacy guarantee will be loose. Again, I understand this is the benchmark from previous works, but still want to express my concern here.

**Summary Of The Paper:**

This paper trains tailored diffusion models with differential privacy. Two intuitive motivations for using diffusion models for DP data generation are 1) the loss function is a simple and scalable $L_2$ regression loss; 2) the denoiser network in diffusion models is simpler and smoother because it is not designed to generate data in a one-shot fashion. Several novel modifications on the design choices of diffusion models are proposed to boost the performance. One worth mentioning modification to the objective function of non-private diffusion models is *noise multiplicity*. At each step, the authors sample $K$ noises and take the average of the gradients of $K$ denoising losses. Experiments show the proposed method achieves SOTA performance.

**Summary Of The Review:**

I recommend acceptance because this paper gives a high-quality implementation of DP diffusion models and demonstrates promising results. I hope the authors could reply to my first concern.

---

> ### Author Response · Authors · 2022-11-15
> **Thank you for questions and feedback**
>
> We thank the reviewer for their feedback.  We are glad that the reviewer finds our work well-written and enjoyed reading it. We are also very pleased that the reviewer recognizes the importance of our modifications to non-private DMs. Below we address specific questions and comments.
>
> - **Additional Experiments:**  We followed the reviewers suggestions and ran additional experiments on more diverse datasets, namely, CIFAR-10 and ImageNet, and we appreciate that the reviewer already acknowledges the difficulty of this setup. We show generated samples and FID scores in Section F.1 in our updated manuscript.
>
>     While it is difficult to recognize classes in the generated images, the model seems to learn approximately correct image statistics, and visually outperforms prior work (on CIFAR-10) by very large margins (see Figure 15). The DP-MERF baseline for the same privacy setting essentially collapses entirely. To the best of our knowledge, other works did not even attempt to generate data from these challenging datasets with rigorous DP guarantees and without the use of auxiliary public data. Consequently, we believe that our DPDM represents a major step forward. We thank the reviewer for suggesting these experiments and hope that they can motivate future research on DP image generative modeling on challenging multimodal datasets.
>
> - **Concern about CelebA:** We thank the reviewer for bringing this point to our attention. To accommodate for the multiple per-identity samples in the dataset, one could potentially introduce the notion of Group Privacy. However, as the reviewer already acknowledges, the CelebA dataset is merely used as a benchmark and we leave an implementation of Group Privacy in DPDMs to future research. Nevertheless, we explicitly added a discussion about this point in the new Section F.3 of our appendix. Upon acceptance, we will rearrange the paper and include this discussion into the main text.

---

> > ### Comment · Reviewer_vz3p · 2022-11-16
> > **Response**
> >
> > Thank you for the new experiments. It's good to see results on both CIFAR-10 and ImageNet. I have no further questions.

---

### Official Review · Reviewer_F4hz · 2022-10-24

**Confidence:** 3
**Correctness:** 4
**Technical Novelty And Significance:** 2
**Empirical Novelty And Significance:** 2
**Recommendation:** 5

**Clarity, Quality, Novelty And Reproducibility:**

The paper is well-structured and clearly written. The definition of the statistical method is easy to follow.
The novelty of the paper is somewhat incremental.
The reproducibility of the paper seems good as relevant details as listed in the paper.

**Strength And Weaknesses:**

Strength:
1. This paper is well-organized and easy to follow.
2. The experimental results on MNIST, Fashion-MNIST, and CelebA datasets validate the effectiveness of the proposed method, and DPDM achieves state-of-the-art performance.

Weaknesses:
1. This paper seems simply adopt the well-established DP-SGD into Diffusion Model training. The novelty seems limited.
2. The motivation of the DPDM is somewhat weak, there exist various generative models, such as GAN, VAE, Diffusion, and so on. Training a GAN is harder than VAE and Diffusion model. However, previous works [G-PATE, PATE-GAN, DPGAN, GS-WGAN] demonstrate the effectiveness of training a DP-GAN-like model.
3. The datasets adopted have small resolutions. For example, the authors use $28 \times 28$ resolution for MNIST and Fashion-MNIST, $32 \times 32$ resolution for CelebA dataset. Training a high-resolution model might be realistic in the application, as the Diffusion Model is well-known for its outstanding performance in high-resolution image synthesis. It might be hard for DP-SGD to train with high resolution while perverse the utility of the outputs.


**Summary Of The Paper:**

This paper studied the differential privacy preservation for the generative model and proposed incorporating the DP-SGD strategy into Diffusion Model training.

**Summary Of The Review:**

The main concern of the reviewer is the novelty of DPDM. It seems that the DPDM is a combination of DP-SGD and DDIM models.

---

> ### Author Response · Authors · 2022-11-15
> **Thank you for questions and feedback (1)**
>
> We thank the reviewer for their feedback. We are glad that the reviewer considers our work well-organized and easy to follow and our experimental, state-of-the-art, performance compelling. Below we address specific questions and comments.
>
> - **Limited Novelty with respect to DP-SGD and Diffusion Models:** We believe that our work connects very important ideas that have been individually very relevant in their respective applications. On the one hand, DP-SGD is arguably the most widely used and most successful method for training neural networks with differential privacy (DP). On the other hand, Diffusion Models (DMs) have become the new state-of-the-art method in the image generative modeling literature, largely outperforming generative adversarial networks (GANs). We also believe that studying DP for DMs is very timely, given the current trend of training very large DMs on internet-scraped data [1,2,3], which has sparked controversy due to the existence of private, sensitive, or proprietary images in their training data. So despite its simplicity, we believe this line of research is very relevant.
>
>     We also want to re-iterate the importance of our novel *noise multiplicity* mechanism: a modification of DP-SGD that is specifically tailored towards training of DPDMs, and which empirically shows large boosts in performance. Furthermore, we also study design choices (e.g. noise schedule and neural network size) and training recipes (train for many epochs, use EMA, use small clipping constants, etc.) necessary to make DMs work well under DP. On top of that, we also provide insights on why DMs are so well suited for DP generative modeling, thereby providing a possible explanation for our strong quantitative results (Section 3.1). Finally, for the first time we show that DP generative models perform on par with task-specific DP classifiers trained on downstream discriminative tasks (Section 5.1). Overall, our state-of-the-art (we outperform all baselines by large margins) experimental results are empirically novel and significant despite the "simplicity" of our method.
>
>     Lastly, we emphasize that we consider the "simplicity" of our method to be a "feature" rather than a "bug". In particular, we believe researchers should build powerful tools that can be easily used by practitioners. Given the "simplicity" as well as our state-of-the-art performance, we believe we lay the foundations towards such a tool for DP generative modeling.
>
> - **Motivation for DPDMs and other Generative Models:** The challenge of training DP-GANs was, in fact, one of the motivations for DPDMs (see the first paragraph in Section 3.1). Generally, when aiming to train a DP generative model, we argue that one should choose the generative modeling framework that is best suited for this task. Previously, as pointed out by the reviewer, most works relied on GAN-based method. However, that is precisely our point: Stepping away from the GAN framework and using a Diffusion Model approach, which has never been done before in DP generative modeling, we can outperform all previous GAN-based DP generative models by large margins. This implies that DP-GANs are, in fact, not "effective" compared to our DPDMs. Our approach leads to empirical results that outperform complex GAN-based setups by large margins across all experiments.
>
>     We want to conclude this discussion by quoting the reviewer: "Training a GAN is harder than ... [a] Diffusion model" – we argue that this is, in fact, very strong motivation to avoid GANs and rather use Diffusion Models. As already discussed above, we believe that as a community we should build tools that are robust to use in practice, and therefore we believe that the community can heavily benefit from our DPDMs.

---

> > ### Author Response · Authors · 2022-11-15
> > **Thank you for questions and feedback (2)**
> >
> > - **DPDMs for high resolution image generation**: We agree with the reviewer that **non-private DMs with millions of parameters** are excelling at high-resolution image synthesis. As the reviewer already suggested, preserving the utility of private DMs for high-resolution image synthesis is very challenging, since it generally requires very large (number of parameters) models. As explained in Section 3.2 (paragraph Neural networks sizes), the $L_2$-norm of the noise added in the DP-SGD update scales linearly to the number of parameters. Therefore, high-resolution DP image generative modeling is very challenging.
> >
> >     That being said, we followed the reviewer's suggestion and tried to learn a model for CelebA at resolution 64x64; see new Section F.2 in the updated manuscript. To accommodate the higher resolution, we added an additional upsampling/downsampling layer to the U-Net, which results in roughly 11% increase in the number of parameters. We find that our generated images at resolution 64x64 (Figure 16, (a)) look more distorted and less sharp than the images at 32x32 resolution (Figure 5). However, we can still clearly recognize faces. In comparison, the only other method that attempted DP synthesis of CelebA at resolution 64x64 is DataLens–our outputs, despite the artifacts, are still orders of magnitudes better. No other works in DP generative modeling even tried 64x64 resolution images synthesis, as this is so challenging with DP guarantees.
> >
> >     We consider our work as an initial step towards high quality DP image generative modeling. On MNIST and Fashion-MNIST, we propose the first DP generative model that can generate images that are practically indistinguishable from non-DP generated images. We hope that our work can motivate further research that potentially down the road enables high-resolution DP image generation.
> >
> >     The reviewer also briefly mentioned utility. We would like to point out that in our other experiments we generally demonstrate very high utility of our DP generative models when the synthesized data is used to train downstream discriminative networks. In fact, as discussed above, for the first time we show that DP generative models perform on par with task-specific DP classifiers trained on downstream discriminative tasks (Section 5.1).
> >
> > **If our reply is satisfactory and the reviewer's questions have been successfully addressed, we would like to kindly ask the reviewer to consider raising their score accordingly. Otherwise, we will be happy to further discuss.**
> >
> > [1] Hierarchical Text-Conditional Image Generation with CLIP Latents. Ramesh et al. 2022.
> >
> > [2] Photorealistic Text-to-Image Diffusion Models with Deep Language Understanding. Saharia et al. 2022.
> >
> > [3] High-Resolution Image Synthesis with Latent Diffusion Models. Rombach et al. 2021.

---

> > > ### Comment · Reviewer_F4hz · 2022-11-17
> > > **Response**
> > >
> > > Thanks for your detailed feedback! The results of high-resolution experiments are interesting. The proposed DPDM outperforms DataLens with a more realistic image synthesis, and the reviewer agrees that the high-quality DP image generative modeling is a challenging task, and DPDM gives an initial attempt that can inspire further research.
> > > The reviewer also agrees that the empirical novelty of the paper after incorporating the higher resolution experiment ($32 \times 32$ to $64 \times 64$) is non-trivial. However, the reviewer still thinks that a simple combination of DP-SGD and the DDIM models may impact the novelty of the overall paper. Therefore, the reviewer holds a borderline recommendation for the paper.

---

> > > > ### Author Response · Authors · 2022-11-17
> > > > **Thank you for your response**
> > > >
> > > > We would like to thank the reviewer for their reply, for acknowledging the challenging nature of DP generative modeling, and for appreciating our additional experiments and the empirical novelty of the paper. We would like to re-iterate, though, that from our perspective simplicity should not imply a lack of novelty. In fact, we think that simplicity is often beneficial, because it implies that the approach can be easily adopted by the community and therefore enables real-world impact. To conclude, let us briefly summarize our main contributions again:
> > > >
> > > > **Technical novelties:**
> > > > - We are the first to propose a method to train differentially private diffusion models.
> > > > - We propose noise multiplicity, a modification of DP-SGD specifically tailored towards differentially private training of diffusion models. We empirically verify that noise multiplicity significantly boosts the performance of our models and reduces the gradient variance of the denoising score matching objective (Section F.4).
> > > > - We provide insights on why diffusion models are well suited for differentially private generative modeling, thereby providing a possible explanation for our strong quantitative results (Section 3.1).
> > > >
> > > > **Empirical novelties:**
> > > > - First and foremost, we outperform all existing differentially private generative models **by large margins** on several standard benchmark tests. For example, on MNIST we improve the state-of-the-art FID from **48.4 to 5.01** and downstream classification accuracy error from **16.8% to 1.9%**.
> > > > - For the first time in differentially private generative modeling, we show that classifiers trained on DPDM-generated synthetic data perform on par with task-specific DP-SGD-trained classifiers (Section 5.1).
> > > > - We study design choices (e.g. noise schedule and neural network size) and training recipes (train for many epochs, use small clipping constants, etc.) necessary to make diffusion models work well under differential privacy.
> > > >
> > > > We would be happy to further discuss. If the reviewer has suggestions on how to further improve our paper, we would also appreciate hearing those. Thank you!

---

### Official Review · Reviewer_d3JQ · 2022-10-25

**Confidence:** 3
**Correctness:** 3
**Technical Novelty And Significance:** 2
**Empirical Novelty And Significance:** 2
**Recommendation:** 3

**Clarity, Quality, Novelty And Reproducibility:**

The clarity and novelty of this paper are moderate. The experiments could be reproducible with the details provided in this paper.

**Strength And Weaknesses:**

Strengths:

1. Studying differential privacy for the diffusion model is a good initiation.
2.The noise multiplicity idea for a less noisy objective function is new and interesting.
3.This paper also provides positive experiment results using DPDMs.

Weakness

1. The motivation for connecting differential privacy with diffusion models is not well explained.  The arguments in section 3.1 do not fully explain the advantage of diffusion models for differential privacy. The privacy guarantee of DPDMs comes from DP-SGD training which is not new. It would be interesting to show the diffusion model itself benefits privacy.
2. The advantage of DP-SGD for differential private diffusion models is not well supported. The paper claims that the DP generative model is challenging due to injected noise in training. But DPDMs proposed in this paper also uses DP-SGD for training which also injects noise into training. It will be more convincing if the paper can add theoretical or experimental support that DP-SGD injects less noise for DPDMs than for other generative models
3. It will be helpful to have a theoretical or empirical explanation that noise multiplicity reduced the noise injected in the training. It also will be interesting if the paper empirically compares the noise variance in the gradient with different levels of K. Also, the paper can clarify if noise multiplicity is unique to diffusion models or if this idea can be generalized to other generative models.


**Summary Of The Paper:**

This paper introduces differential private diffusion models (SPDMs) and studies DP-SGD for DPDMs training. This paper proposes a modification of the diffusion model training objective called noise multiplicity to boost performance. The DPDMs experiments show improvement in image generation tasks.

**Summary Of The Review:**

I feel the approach proposed in this paper is not well motivated. The significance needs to be improved. Overall, I think this submission is not ready for ICLR. Thus, I recommend rejection.

---

> ### Author Response · Authors · 2022-11-15
> **Thank you for questions and feedback (1)**
>
> We thank the reviewer for their feedback.  We are glad that the reviewer appreciates our initiative to study differentially private (DP) diffusion models (DMs), the novelty of noise multiplicity (our modification of DP-SGD tailored towards the training of DMs) and our positive experimental results.
>
> We believe, however, that the reviewer severely misunderstood some aspects of our work. Most importantly, we want to clarify that attaining differential privacy (DP) in generative models carries significance in itself. Informally, applying DP to generative modeling quantifiably limits the influence of *any* single data point to the model; in other words, it limits how much the model can memorize the training data (more details below). Below we address specific questions and comments.
>
> - **"The privacy guarantee of DPDMs comes from DP-SGD training which is not new":** This is correct: DP-SGD is a well-established method and can be considered as the workhorse of DP training (see our Introduction & Related Work for many citations). We want to emphasize that our work is not about proposing a new DP algorithm, but rather it is the first work to train differentially private diffusion models. We choose to use DP-SGD as a tool to achieve this goal. Similar to our work, DP-SGD has, for example, been used as a tool to train DP GANs [4, 5, 6].
>
>     To reiterate, our novelty lies in the combination of DMs with DP-SGD, modifications of DP-SGD specifically tailored towards the DP training of DMs (noise multiplicity), as well as the study of design choices (e.g. noise schedule and neural network size) and training recipes (train for many epochs, use EMA, use small clipping constants) specifically for DPDMs (and different from non-DP DMs) that greatly influence the performance of DPDMs. We would also like to point to our empirical novelty: We outperform all baselines by huge margins on standard benchmarks and, for the first time, show that downstream discriminative models trained on DPDM-generated data perform on par with task-specific DP-trained discriminative models (see Section 5.1). We additionally provide insights on why diffusion models can potentially be successful for DP training as discussed in Section 3.1.
>
> - **"It would be interesting to show the diffusion model itself benefits privacy":** Similar to GANs, VAEs, and other generative models, DMs in itself do not provide any DP guarantees. DMs learn a denoiser network $D$ that predicts a clean data point (image in our case) $\mathbf{x}$ given a noisy data point $\mathbf{x} + \mathbf{n}$. During standard, non-DP training, of DMs, which effectively amounts to minimizing our Eq. (3), the parameters of the denoiser network are updated using gradients that have direct access to the clean data point $\mathbf{x}$. There is no guarantee that the denoiser does not memorize (overfit to) any particular data point, and therefore DMs do not provide any rigorous DP guarantees by nature.
>
> - **"The motivation for connecting differential privacy with diffusion models is not well explained:"** As stated in the introduction of this rebuttal, we believe that the reviewer misunderstood that attaining DP in generative models carries significance in itself. In the following, we are giving two examples:
>
>     1.) There is a current trend to train very large diffusion models on  Internet-scraped data [1,2,3]. This has sparked controversy due to the existence of private, sensitive, or proprietary images in their training data. Achieving DP in image generators, would, for example, guarantee that the model does not ``steal'' any art by directly overfitting to single data points (for example an image of a painting).
>
>     2.) Modern deep learning usually requires significant amounts of training data. However, sourcing large datasets in privacy-sensitive domains is often difficult. To circumvent this challenge, generative models trained on sensitive data with privacy guarantees that protect the dataset participant can provide access to large synthetic data instead, which can be used flexibly to train different downstream models. In other words, DP generative models can serve as a flexible and powerful data-sharing medium in privacy sensitive domains.
>
>     In conclusion, DP is an essential element in the responsible use of data when training generative models, and advancing the frontier of DP generative models can have major positive societal impact.

---

> > ### Author Response · Authors · 2022-11-15
> > **Thank you for questions and feedback (2)**
> >
> > - **"The arguments in section 3.1 do not fully explain the advantage of diffusion models for differential privacy:"** In 3.1 we lay out additional motivation (to the above) for the specific combination of DP-SGD with DMs. Regarding (i) in 3.1, it is well known that GANs are notoriously hard to optimize and we provide corresponding references. For (ii) Sequential denoising and (iii) Stochastic diffusion model sampling we show empirical evidence (Figure 2 and Sampling Ablation in 5.2, respectively). Although these arguments are no formal proofs, we do believe that they offer a plausible explanation for the very strong numerical performance of DPDMs we observed in our experiments. Generative modeling, as many other fields in machine learning, is greatly driven by empirical results, and we clearly show in our experiments that DPDMs are outperforming existing DP generative models by large margins. The arguments in 3.1 try to offer some insight; however, in the end, we believe the results speak for themselves.
> >
> > - **On the combination of diffusion models and DP-SGD:** The reviewer might have misunderstood DP-SGD and our claim. First of all, the amount (magnitude) of noise injected by DP-SGD is a design parameter; it can be computed a priori given the desired privacy protection level, size of the dataset, batch size, clipping threshold, and number of iterations [7, 8]. Fixing these parameters, *DP-SGD injects noise of the same magnitude regardless of what model (DM, GAN, VAE) is being trained.*
> >
> >     Practically all DP mechanisms require noise injection in some form. Thus, the claim that DP generative learning is challenging is not specific to DP-SGD. *The advantage of DPDMs is that, given the same privacy budget and magnitude of noise injection, they outperform other models that also use DP-SGD, such as GANs, by large margins.* We argue that this is at least partially due to the different advantages of DMs over other generative models, such as GANs, discussed in Section 3.1. Furthermore, we experimentally validate design choices such as noise multiplicity, noise schedule, and sampling strategy, which all contribute to performance uplift and are specifically designed for the DP setting and different from standard non-DP DM training.

---

> > > ### Author Response · Authors · 2022-11-15
> > > **Thank you for questions and feedback (3)**
> > >
> > > - **On the variance reduction of noise multiplicity**: We want to emphasize that there are two sources of noise in training of DPDMs. Firstly, DP-SGD training clips per-example-gradients and noises them. As discussed above, this (amount of) noise is fixed and cannot be modified without breaking DP guarantees. The other source of noise is due to the expectation over three distributions in our learning objective (Eq. (3)). In particular, as is common practice in most disciplines in machine learning, to ensure numerical tractability we replace expectations in training objectives with Monte Carlo estimates. For a generic function $r$ over distribution $p(\mathbf{k})$, we have $\mathbb{E}\_{p(\mathbf{k})}[r(\mathbf{k})] \approx \frac{1}{m} \sum_{i=1}^m r(\mathbf{k}_i)$, where $\\{\mathbf{k}_i\\}\_{i=1}^m \sim p(\mathbf{k})$ (Monte Carlo estimator for expectation of function $r$ with respect to distribution $p$).
> > >
> > >     The Monte Carlo estimate is a noisy unbiased estimator of the expectation $\mathbb{E}\_{p(\mathbf{k})}[r(\mathbf{k})]$ with variance $\frac{1}{m} \mathrm{Var}_p[r]$, where $\mathrm{Var}_p[r]$ is the variance of $r$ itself. This is a well-known fact; see for example Chapter 2 of the excellent book by Owen (https://artowen.su.domains/mc/Ch-intro.pdf).
> > >
> > >     DM training is actually very noisy, as we deal with an expectation not over one distribution but three: 1) the distribution over data points $p(\mathbf{x})$ 2) the distribution over noise levels $p(\sigma)$, and 3) the conditional distribution of noise given the noise level $p(\mathbf{n} \mid \sigma)$. The Monte Carlo estimator over the first distribution cannot be changed, as it depends on the potentially sensitive data and is therefore inherently tied to DP-SGD and a modification would break DP guarantees. However, we are free to modify the Monte Carlo estimator for the joint distribution $p(\sigma, \mathbf{n}) = p(\sigma) p(\mathbf{n} \mid \sigma)$. In particular, with noise multiplicity, we increase the amount of samples $m$ from 1 to $K$, reducing the factor in front of the variance from $1$ to $1/K$. As suggested by the reviewer, we also show this variance reduction empirically by considering the distribution of gradient variances (over the model parameters) for different $K$; the experiment can be found in the new Section F.4 of our updated manuscript.
> > >
> > >     One may wonder why we would not want to apply noise multiplicity to non-DP DMs in the first place. The answer to that is simple: noise multiplicity may generate less noisy gradients; however, it leads to higher computational cost as the large denoiser network needs to be evaluated $m$ times ($K$ in our case). Furthermore, training non-DP DMs longer (more iterations) has a similar effect to noise multiplicty so there is no need for it in the first place. However, as discussed above, the number of iterations in DP training of DMs is limited and additional denoiser calls per iteration are not the bottleneck in the DP setting. Therefore, rather than training for very many noisy iterations, we train with less, more accurate iterations, leveraging our noise multiplicity. As our ablations show, this technical novelty is a crucial ingredient in pushing DPDMs to state-of-the-art performance.
> > >
> > >     Lastly, we are not aware of any other DP generative models for which noise multiplicity would be directly applicable. DMs have a unique characteristic that during training an expectation over three distributions is estimated, two of which can be made less noisy with noise multiplicity.
> > >
> > > We are planning to emphasize some of the points of this discussion in the final paper and we would happily incorporate any further suggestions by the reviewer on how to avoid misunderstandings. **If our reply is satisfactory and the reviewer's questions have been successfully addressed, we would like to kindly ask the reviewer to consider raising their score accordingly. Otherwise, we will be happy to further discuss.**
> > >
> > > [1] Hierarchical Text-Conditional Image Generation with CLIP Latents. Ramesh et al. 2022.
> > >
> > > [2] Photorealistic Text-to-Image Diffusion Models with Deep Language Understanding. Saharia et al. 2022.
> > >
> > > [3] High-Resolution Image Synthesis with Latent Diffusion Models. Rombach et al. 2021.
> > >
> > > [4] Differentially Private Generative Adversarial Networks for Time Series, Continuous, and Discrete Open Data. Frigerio et al. 2019
> > >
> > > [5] DP-CGAN: Differentially Private Synthetic Data and Label Generation. Torkzadehmahani et al. 2019.
> > >
> > > [6] Differentially Private Generative Adversarial Network. Xie et al. 2018
> > >
> > > [7] Rényi differential privacy. Mironov. 2017
> > >
> > > [8] Rényi Differential Privacy of the Sampled Gaussian Mechanism. Mironov et al. 2019.

---

### Author Response · Authors · 2022-11-15
**Comment to all Reviewers**

We thank all reviewers for engaging in the review process and apologize for the late reply. We hope that we can engage in active discussions with the reviewers over the next days.

In our rebuttal replies, we attempted to address specific questions and comments as clearly and detailed as possible. All our additions to the paper are currently placed in App. F due to space limitations. If the paper will be accepted, we will re-organize the content and potentially bring some of the context from the appendix into the main paper.

---

### Decision · Program_Chairs · 2023-01-20

**Decision:**

Reject

**Justification For Why Not Higher Score:**

The paper novelty is limited and the evaluation on the privacy of the resulting models/ synthetic data is not provided.

**Justification For Why Not Lower Score:**

N/A

**Metareview: Summary, Strengths And Weaknesses:**

The paper proposes to use DP-SGD for training the score function in training diffusion models (DM), and proposes noise multiplicity to improve the SNR in the cost  estimation for training the DM.

The main strength in the paper is that shows on image datasets that the DP models quality is improved with respect to previous work.

The main weaknesses of the work is the  novelty of  the work, as it  builds on DP-SGD and noise multiplicity previously proposed in De et al. Another weakness is that the evaluation is only on image quality / accuracy of downstream task on DP synthetic data and there is not any metric on the privacy of the obtained models.



**Summary Of Ac-Reviewer Meeting:**

N/A